# Blockade of VEGF-C signaling inhibits lymphatic malformations driven by oncogenic *PIK3CA* mutation

Ines Martinez-Corral[1,9], Yan Zhang[1,9], Milena Petkova [1], Henrik Ortsäter[1], Sofie Sjöberg[1], Sandra D. Castillo [2], Pascal Brouillard [3], Louis Libbrecht[4], Dieter Saur[5], Mariona Graupera[2], Kari Alitalo[6], Laurence Boon[3,7], Miikka Vikkula [3,8] & Taija Mäkinen[1✉]

Lymphatic malformations (LMs) are debilitating vascular anomalies presenting with large cysts (macrocystic) or lesions that infiltrate tissues (microcystic). Cellular mechanisms underlying LM pathology are poorly understood. Here we show that the somatic *PIK3-CA[H1047R]* mutation, resulting in constitutive activation of the p110α PI3K, underlies both macrocystic and microcystic LMs in human. Using a mouse model of *PIK3CA[H1047R]*-driven LM, we demonstrate that both types of malformations arise due to lymphatic endothelial cell (LEC)-autonomous defects, with the developmental timing of p110α activation determining the LM subtype. In the postnatal vasculature, *PIK3CA[H1047R]* promotes LEC migration and lymphatic hypersprouting, leading to microcystic LMs that grow progressively in a vascular endothelial growth factor C (VEGF-C)-dependent manner. Combined inhibition of VEGF-C and the PI3K downstream target mTOR using Rapamycin, but neither treatment alone, promotes regression of lesions. The best therapeutic outcome for LM is thus achieved by co-inhibition of the upstream VEGF-C/VEGFR3 and the downstream PI3K/mTOR pathways.

[1] Uppsala University, Department of Immunology, Genetics and Pathology, Dag Hammarskjölds väg 20, 751 85 Uppsala, Sweden. [2] Vascular Signaling Laboratory, Institut d´Investigació Biomèdica de Bellvitge (IDIBELL), 08908L´Hospitalet de Llobregat, Barcelona, Spain. [3] Human Molecular Genetics, de Duve Institute, University of Louvain, Brussels, Belgium. [4] Center for Vascular Anomalies, Division of Pathology, Cliniques universitaires Saint Luc, University of Louvain, 10 avenue Hippocrate, B-1200 Brussels, Belgium. [5] Department of Internal Medicine 2, Klinikum rechts der Isar, Technische Universität München, Ismaningerstr. 22, 81675 München, Germany. [6] Wihuri Research Institute and Translational Cancer Biology Program, Biomedicum Helsinki, FIN-00014 University of Helsinki, Helsinki, Finland. [7] Center for Vascular Anomalies, Division of Plastic Surgery, Cliniques universitaires Saint Luc, University of Louvain, 10 avenue Hippocrate, B-1200 Brussels, Belgium. [8] Walloon Excellence in Lifesciences and Biotechnology (WELBIO), University of Louvain, Brussels, Belgium. [9] These authors contributed equally: Ines Martinez-Corral, Yan Zhang. ✉email: taija.makinen@igp.uu.se

Vascular malformations are chronic, often congenital pathologies that can manifest in different types of blood and lymphatic vessels. These diseases commonly arise from abnormalities in the endothelial cells (ECs) of the affected vessel type(s) that lead to structural and functional vascular defects causing deformation, pain, morbidity, and organ dysfunction[1]. Genome sequencing efforts have identified causative mutations for different types of malformations and opened up possibilities for therapeutic intervention specifically targeting the aberrant signal transduction pathways[2,3].

Somatic activating mutations in the *PIK3CA* gene, encoding the p110α catalytic subunit of phosphatidylinositol 3-kinase (PI3K), were identified as causative of ~20% of venous malformations (VM)[4–6], and the majority of lymphatic malformations (LM)[7,8]. The most common VM/LM mutations affecting the helical domain (E542K, E545K) or the kinase domain (H1047R, H1047L) of p110α are identical to those previously found in cancer and other genetic syndromes characterized by tissue overgrowth[9]. Both types of mutations result in basal activation of the PI3K pathway by enhancing dynamic events in the natural activation of p110α that lead to increased lipid binding[10].

The PI3K lipid kinases control a variety of cellular functions and developmental and homeostatic processes in response to extracellular signals by regulating the plasma membrane phosphatidylinositol (3,4,5)-triphosphate ($PIP_3$) levels[11]. Of the four p110 isoforms, the ubiquitously expressed p110α has emerged as the key downstream effector of growth factor receptor signaling in most cell types and in particular in the endothelium. Genetic loss-of-function studies in mice demonstrated an important role of p110α in the development of both blood and lymphatic vessels[12–14]. Conversely, conditional expression of the *PIK3CA*-activating mutation in endothelial cells led to vascular overgrowth and malformations in mice[4,5,15,16]. Identification of *PIK3CA* mutations as drivers of vascular malformations has opened up a possibility for the therapeutic use of PI3K inhibitors in these diseases. Rapamycin and its analogues (sirolimus, everolimus) that target the PI3K downstream effector mTOR can stop the progression of vascular malformations in mice and human, and improve the patients' quality of life[3,5,15–19]. However, regression of lesions is observed only in a minority of patients[3], which calls for a need to develop new more effective therapies.

Compared with the malformations affecting the blood vasculature, LMs have received less attention despite often severe complications for patients. LMs are characterized by large fluid-filled cysts (macrocystic LM), or diffuse, infiltrative lesions sometimes consisting of small vesicles containing lymph or blood (microcystic LM)[19,20]. Many patients show a mixed phenotype with a combination of large and small cysts. Lesion growth may be progressive and, depending on the location, result in severe complications such as infections and impairment of breathing or swallowing. Macrocystic LM can be usually effectively treated with sclerotherapy or surgical resection. By contrast, the treatment of microcystic LM is challenging due to their infiltrative growth, and curative therapies are currently lacking.

Here we studied the pathophysiological mechanisms of *PIK3CA*-driven LM. We show that both macrocystic LM and microcystic LM are driven by the activating *PIK3CA*[H1047R] mutation, with the developmental timing of activation of the p110α PI3K signaling in lymphatic endothelia determining the LM subtype. We further show that the growth of *PIK3CA*[H1047R]-driven microcystic LM in mice is dependent on the upstream lymphangiogenic vascular endothelial growth factor C (VEGF-C)/VEGFR3 signaling. Combined inhibition of VEGF-C signaling and the PI3K downstream target mTOR using Rapamycin, but neither treatment alone, promotes the regression of experimental LM in mice. Our findings provide important implications for the

treatment of microcystic LM in human and suggest that therapies targeting the key upstream pathway in combination with PI3K pathway inhibition may be relevant for other *PIK3CA*-driven pathologies.

## Results

### *PIK3CA*[H1047R] mutation underlies micro- and macrocystic LMs.
To address whether the two subtypes of LM require different *PIK3CA* mutations, potentially driving different cellular responses, or if the same mutation can underlie both microcystic and macrocystic LMs, we focused on patients with a somatic *PIK3CA*[H1047R] mutation. Clinical features of five patients selected for the study are summarized in Table 1. Histologic features of the lesions were investigated using tissue sections from the patients to confirm lymphatic identity of the lesions and the LM subtype (Fig. 1a, b, Supplementary Fig. 1a).

Lesions from two patients with macrocystic LM were characterized by convoluted structures with large lumens, which in some cases contained lymph, lymphocytes and/or red blood cells (RBCs) (Fig. 1a). The endothelium lining the cystic lumens was positive for PDPN and PROX1 (Supplementary Fig. 1a), confirming lymphatic endothelial identity. LYVE1 (Supplementary Fig. 1a) and VEGFR3 (Fig. 1b, Supplementary Fig. 1b) staining was weak or lacking in the largest cysts, in particular in areas surrounded by αSMA+ smooth muscle cells. High VEGFR3 expression was instead observed in small-caliber lymphatic vessels in areas within the cyst wall containing infiltrated CD45+ cells (Fig. 1b). Microcystic LMs were characterized by smaller cysts and vessels with irregular lumens and no smooth muscle cell layer (Fig. 1b). The endothelium of microcystic LMs was positive for VEGFR3 (Fig. 1b), PDPN and PROX1 (Supplementary Fig. 1a). LYVE1 staining was patchy, similarly as in macrocystic LMs (Supplementary Fig. 1a). Similar to the macrocystic LM, and in agreement with previous reports[21], a large number of CD45+ cells (Fig. 1b) and presence of lymphoid aggregates (Supplementary Fig. 1c) were also observed in all three microcystic LMs.

These data show that the two distinct LM subtypes showing different morphological and molecular features result from the same somatic mutation in *PIK3CA*.

### Developmental timing of PI3K activation determines LM subtype.
To investigate if both macrocystic and microcystic *PIK3CA*-driven LMs are caused by lymphatic endothelial cell (LEC)-autonomous defects, we generated a mouse model that allows inducible LEC-specific activation of the causative mutation *PIK3CA*[H1047R]. Mice carrying a knock-in *PIK3CA*[H1047R] allele in the *Rosa26* locus, silenced by a lox-stop-lox cassette until Cre exposure[22], were crossed with the *Vegfr3-CreER*[T2] mice expressing tamoxifen-inducible Cre recombinase specifically in lymphatic endothelia[23] (Fig. 2a). Since LMs are often present at birth, we first induced activation of *PIK3CA*[H1047R] expression at the early stage of lymphatic development by administering 4-hydroxytamoxifen (4-OHT) to pregnant females at embryonic day (E)11 (Fig. 2a). Whole-mount immunofluorescence of the skin of embryos at E17 revealed two types of lesions in the lymphatic vasculature. The majority of lesions appeared as large isolated cysts that were localized mainly to the cervical, and less frequently to the sacral region of the skin (Fig. 2b, c, Supplementary Fig. 2a, b). In addition, areas of lymphatic vessel hypersprouting were observed in the thoracic region of the skin (Fig. 2b). Lymphatic endothelium showed normal identity and expression of LEC markers including PROX1 (Fig. 2b, Supplementary Fig. 2b). Histological examination of the skin revealed that the cysts consisted of a single lumen (Fig. 2d) that frequently contained RBCs (Supplementary Fig. 2b, c). The morphological

**Table 1 Clinical features of patients with LM driven by H1047R mutation in *PIK3CA*.**

| Case | LM type | Gender | Onset of disease | Symptoms | Treatment |
|---|---|---|---|---|---|
| 1 | Microcystic | M | 10 y | Localized soft tissue hypertrophy and dermal lymphatic vesicles causing oozing on the right flank | Surgical resection at 13 years of age |
| 2 | Microcystic | M | Congenital | Right hemifacial cheek hypertrophy; lesion invading the facial nerve and the parotid gland; intrabuccal mucosal small vesicles causing daily oozing | Surgical resection at 2 years of age |
| 3 | Microcystic | M | Congenital | Born with a small discoloration on his right hypothenar area that slowly indurated. By the age of 2 years, an ill-defined 5 × 5 cm diameter mass appeared progressively with time; pain on palpation | Surgical resection at 6 years of age |
| 4 | Macrocystic | M | Congenital | Lesion on the right neck and lower cheek; increased in size during chickenpox infection at 19 months of age | Sclerotherapy, intralesional injection of picibanyl at 22 months of age; surgical removal at 27 months of age |
| 5 | Macrocystic | F | Congenital | Lesion on the right neck englobing the parotid gland; increased in size at 15 days of life and at 1.5 years of age with important intralesional bleeding | Intralesional injection of picibanyl at 3 and 4 years; surgical removal at 7 years due to frequent infections and increase in volume |

features of the lesions and their predominant localization to the neck region of the skin were thus reminiscent of human macrocystic LM.

Next, we induced activation of $PIK3CA^{H1047R}$ expression in LECs during late embryonic (Supplementary Fig. 2a) or early postnatal development (Fig. 2a). Administration of 4-OHT to pregnant females at E14 led to increased lymphatic vessel branching in all regions of the dorsal skin of E17 $PIK3CA^{H1047R}$;$Vegfr3$-$CreER^{T2}$ embryos, but no macrocystic lesions were observed ($n = 6$ embryos) (Supplementary Fig. 2d, e). Similarly, analysis of the vasculature in the ear skin of neonatally induced mice revealed well-defined malformations that were characterized by a dense network of hyperbranched lymphatic vessels (Fig. 2e). These lesions further developed into large sheet-like vascular structures that frequently showed bleeding in the lesion area (Fig. 2e). A similar lymphatic hypersprouting phenotype was observed using another LEC-specific Cre line, the $Prox1$-$CreER^{T2}$[24] (Supplementary Fig. 3a–c). Histological examination of the lesions in the $PIK3CA^{H1047R}$;$Prox1$-$CreER^{T2}$ ear showed multiple lumens (Fig. 2f) and presence of RBCs both inside and outside of the lymphatic vessels (Supplementary Fig. 3d), reminiscent of microcystic LMs in human patients (Fig. 1b).

In summary, these results demonstrate that developmental timing of activation of the p110α signaling in LECs determines the LM subtype. Early embryonic activation leads predominantly to formation of lesions that recapitulate features of human macrocystic LM, whereas activation during late embryonic or neonatal development results in microcystic LM (Supplementary Table 1). Our data further suggest that the physical location of the lesion also impacts on their characteristics, with most macrocystic lesions developing in the cervical region of the skin.

**$PIK3CA^{H1047R}$ expression confers migratory phenotype in LECs.** We focused our investigation on microcystic LMs that are difficult to treat due to their infiltrative pattern of growth. To allow analysis of the early steps of microcystic LM development, we analyzed the acute response of postnatal ear vasculature of $PIK3CA^{H1047R}$;$Vegfr3$-$CreER^{T2}$ mice 10 days after neonatal 4-OHT administration (Fig. 3a). We observed formation of new vessels showing a typical 'spiky' appearance of active vessel sprouts (Fig. 3b). Quantification of vessel ends and their morphology demonstrated blunted morphology of most lymphatic capillaries and only a few active sprouts in the ear skin of 3-week-old mice (Fig. 3b), in agreement with previous observations[25].

$PIK3CA$ mutant vasculature showed an increase in the number of vessel ends, the majority of which had the spiky morphology of active sprouts (Fig. 3b). Staining for phosho-S6 as a readout of PI3K activity[26] revealed signal selectively in active lymphatic sprouts in the mutant vasculature, similar to lymphangiogenic sprouts of embryonic vessels (Fig. 3c). LECs within the active sprouts also displayed straight zipper-like cell-cell junctions, a hallmark of lymphangiogenic vessels[27], as opposed to oak-leaf shaped LECs of normal capillaries that possess button-like junctions (Fig. 3d).

To study more advanced lesions, we utilized models permitting robust and reproducible induction of localized vs. generalized $PIK3CA^{H1047R}$-driven response in the postnatal ear skin using the $Prox1$-$CreER^{T2}$ and $Vegfr3$-$CreER^{T2}$ models. Topical application of a low dose of 4-OHT (0.5-2 μg) into one ear of a $PIK3CA^{H1047R}$;$Prox1$-$CreER^{T2}$ mouse induced locally restricted Cre recombination and lesions in the treated ear (Supplementary Fig. 3a, b). A similar sprouting response and progression of lesions was observed as previously seen in the $Vegfr3$-$CreER^{T2}$ model (Supplementary Fig. 3c, d). In contrast, topical administration of a higher dose of 4-OHT (100 μg) into one ear of a $Vegfr3$-$CreER^{T2}$ mouse led to systemic Cre-mediated recombination (Supplementary Fig. 4a, b), resulting in a generalized $PIK3CA^{H1047R}$-driven sprouting response of the entire dermal lymphatic vasculature (Supplementary Fig. 4c, d). Progressive increase in lymphatic vessel hyperbranching (Supplementary Fig. 4e) was associated with increase in the number of active sprouts (Supplementary Fig. 4f), as well as in LEC proliferation (Supplementary Fig. 4g) and numbers (Supplementary Fig. 4h).

To characterize the LEC phenotype associated with p110α activation, we isolated primary dermal LECs from the $PIK3CA^{H1047R}$;$Vegfr3$-$CreER^{T2}$ mice by sequential selection of PECAM1$^+$ ECs and LYVE1$^+$ LECs, as previously described[23]. Staining for VE-cadherin and PROX1 demonstrated high purity of the isolated LEC population (Supplementary Fig. 5a). $PIK3CA^{H1047R}$-expressing LECs, treated in vitro with 4-OHT, showed reorganization of cell-cell junctions and cytoskeleton characterized by increase in serrated junctions with VE-cadherin fingers, and increased actin stress fiber formation compared with untreated controls (Fig. 3e). Such changes in junctional morphology have previously been linked to active EC migration and vessel sprouting[28–30]. To test if the observed morphological changes correlated with increased migratory behavior, we exposed $PIK3CA^{H1047R}$-expressing and control LECs to scratch wound assay. Live imaging of scratch wounded cell monolayers

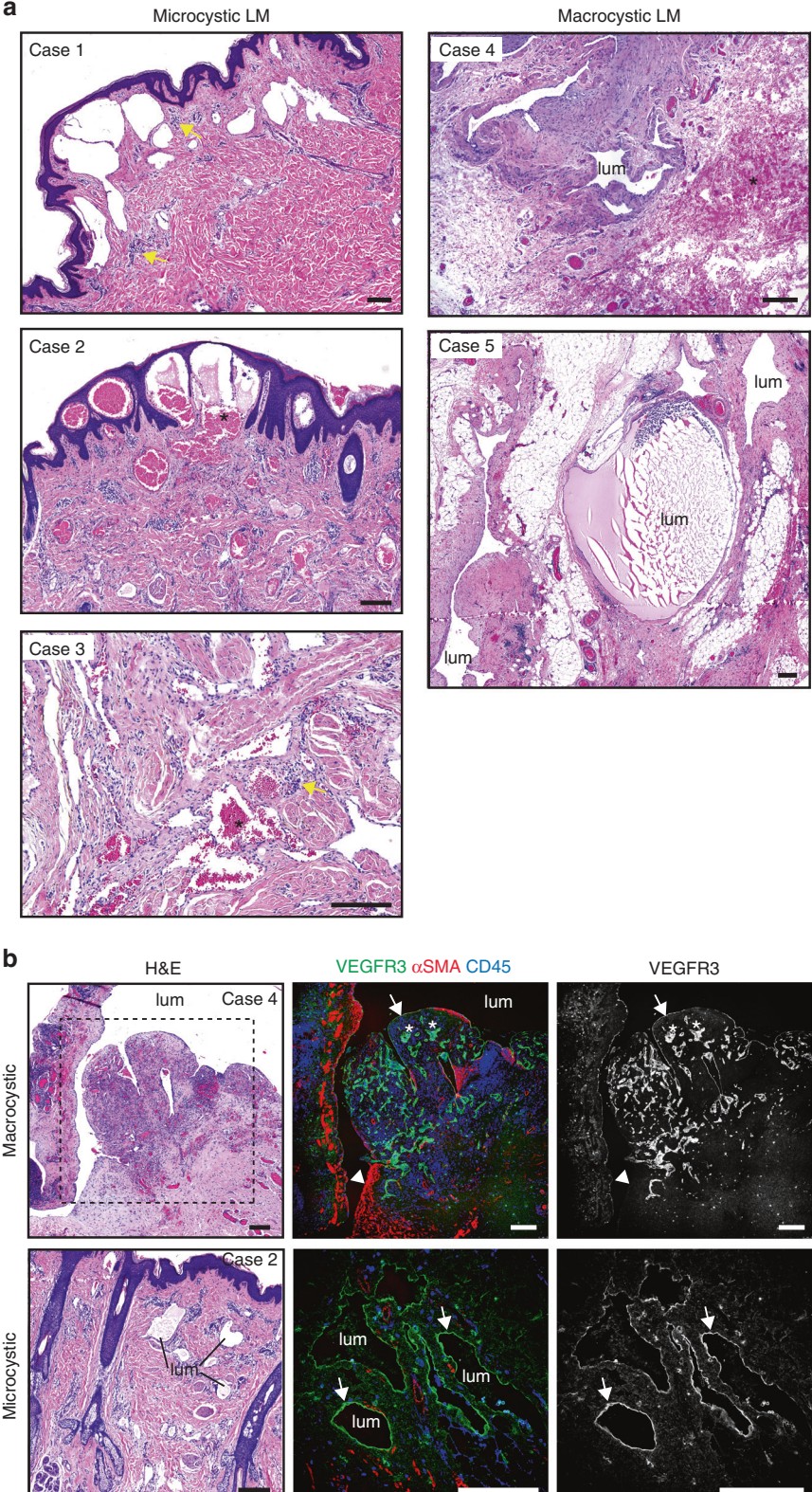

**Fig. 1 PIK3CA mutations underlie both micro- and macrocystic LM. a** Hematoxylin and eosin stained sections of three microcystic (on the left) and two macrocystic (on the right) LMs. Note RBCs inside and outside of the malformations (asterisks), and lymphoid cell infiltration (yellow arrows). **b** On the left: hematoxylin and eosin stained sections of macrocystic and microcystic LM. On the right: immunofluorescence staining showing VEGFR3 expression in the endothelium lining both macrocystic and microcystic LM (arrows), except for areas associated with smooth muscle cells (arrowhead). Boxed area is shown in both hematoxylin and eosin staining and immunofluorescence staining. Note high expression of VEGFR3 in small-caliber lymphatic vessels in areas containing infiltrated CD45[+] cells (asterisks). lum = vessel/cyst lumen. Images in (**a**, **b**) are representative of $n = 3$ microcystic LM and $n = 2$ macrocystic LM lesions. Scale bars: 200 μm (**a**, **b**).

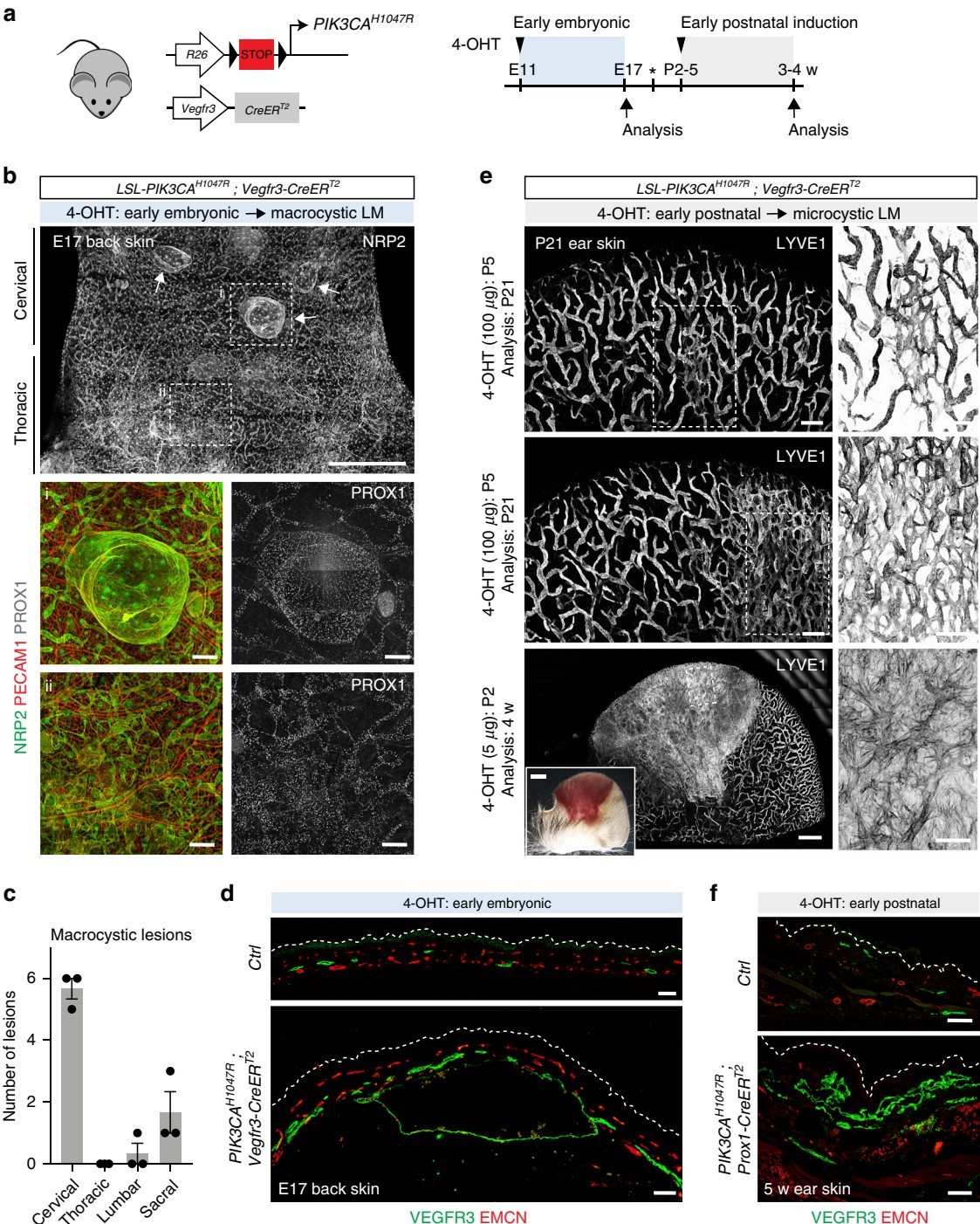

**Fig. 2 Developmental timing of activation of *PIK3CA* determines LM subtype. a** Genetic constructs and experimental plan for tamoxifen-inducible activation of *PIK3CA^{H1047R}* expression in lymphatic endothelia at early embryonic (blue, 4-OHT 2 mg) or postnatal (gray, 4-OHT dose as indicated) development. Asterisk indicates the date of birth. **b–d** Characterization of *PIK3CA^{H1047R}*-driven macrocystic LM. Whole mount immunofluorescence of back skin from an E17 *PIK3CA^{H1047R};Vegfr3-CreER^{T2}* embryo (**b**). Antibodies and regions of skin are indicated, and boxed areas (i, ii) are magnified below. Arrows point to cyst-like malformations in the neck (cervical) region of the skin that were quantified in (**c**). Data represent mean ± s.e.m. Immunofluorescence analysis of paraffin-embedded sections of lymphatic lesions showing overgrowth of lymphatic (VEGFR3+) but not blood vessels (EMCN+) (**d**).
**e**, **f** Characterization of *PIK3CA^{H1047R}*-driven microcystic LM. Immunofluorescence analysis of skin whole-mounts (**e**; *Vegfr3-CreER^{T2}* model) or paraffin sections (**f**; *Prox1-CreER^{T2}* model), and boxed areas (**e**) are magnified. 4-OHT dosage, timing of administration and analysis are indicated. Inset in (**e**) shows bleeding in the lesion area. Dotted lines in (**d**, **f**) indicate epidermis. Scale bars: 50 μm (**d**, **f**), 250 μm (**b** (i, ii), **e**), 2 mm (overview images in **b**, **e**). Source data are provided as a Source Data file.

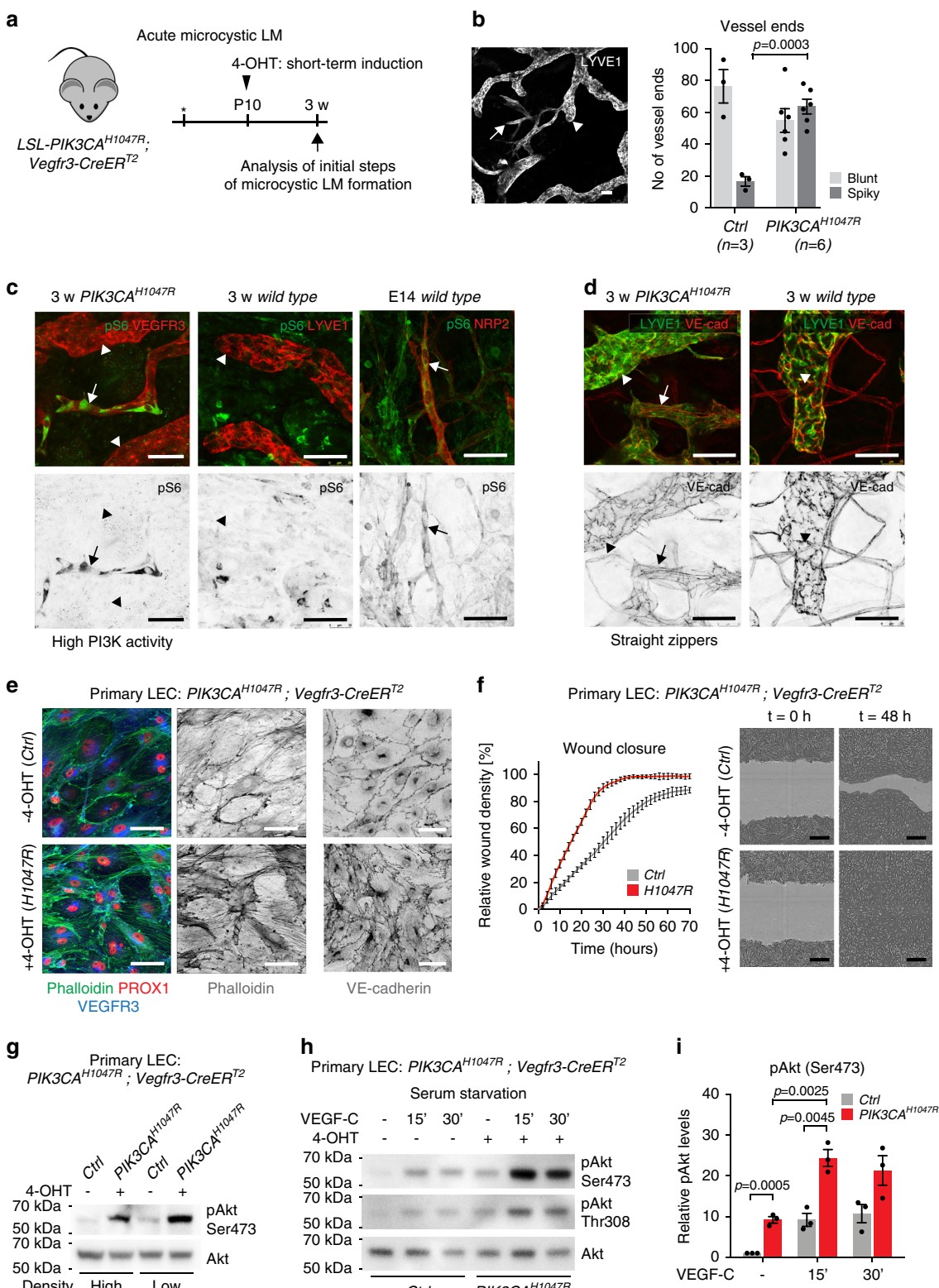

showed enhanced wound closure of 4-OHT-treated LECs expressing the constitutive active p110α compared with vehicle-treated control LECs (Fig. 3f). No further increase in the speed of wound closure in the $PIK3CA^{H1047R}$-expressing primary LECs was observed in the presence of the specific lymphangiogenic growth factor VEGF-C (Supplementary Fig. 5b), suggesting that the change in the cellular phenotype represents a cell-autonomous effect of oncogenic PI3K activity on the LEC.

Together, these results demonstrate that p110α activation confers a migratory LEC phenotype and induces lymphatic vessel sprouting that leads to the hyperbranched network of vessels defining LM lesions.

**VEGF-C induces hyperactivation of Akt in $PIK3CA^{H1047R}$-LECs.** Next, we studied the effect of p110α activation on

**Fig. 3 Migratory phenotype and VEGF-C-induced hyperactivation of Akt in *PIK3CA^H1047R*-expressing LECs. a** Experimental plan for short-term 4-OHT induction for the analysis of initiation of microcystic LM formation in *PIK3CA^H1047R;Vegfr3-CreER^T2* mice. Asterisk indicates the date of birth. **b** Whole-mount immunofluorescence of ear skin of 4-OHT-treated (100 μg) P21 *PIK3CA^H1047R;Vegfr3-CreER^T2* mice (on the left) and quantification of the morphology of vessel ends in control (*n* = 3) and *PIK3CA^H1047R* (*n* = 6) mice (mean ± s.e.m., Two-tailed unpaired Student's *t*-test) (on the right). Mutant shows increase in spiky vessel ends representing active sprouts (arrow) in comparison to blunt ends of normal capillaries (arrowhead). **c** Staining for phospho-S6 staining as a readout of PI3K activity, showing high signal in active lymphatic vessel sprouts (arrows) in 3-week-old mutant skin and embryonic back skin. No signal was detected in normal capillaries (arrowheads). **d** Staining for VE-cadherin showing straight zipper-like junctions, typical of lymphangiogenic vessels, in active lymphatic sprouts (arrows) in mutant skin as opposed to oakleaf shaped LECs with button junctions in normal capillaries (arrowhead). **e** Immunofluorescence of primary LECs isolated from *PIK3CA^H1047R;Vegfr3-CreER^T2* mice, showing increase in stress fiber formation and disruption of cell-cell junctions in 4-OHT-treated (*PIK3CA^H1047R*-expressing) compared with vehicle-treated (*Ctrl*) cells. Images are representative of three independent experiments. **f** IncuCyte scratch assay showing faster wound closure in 4-OHT-treated (*PIK3CA^H1047R*-expressing) compared with vehicle-treated (*Ctrl*) primary LECs. Data is representative of two independent experiments showing mean relative wound density (*n* = 7 (*Ctrl*) or 5 (*H1047R*) wells ± s.d.). The difference between area under the curves: *P* < 0.0001 (Two-tailed unpaired Student's *t*-test). Representative images of wells at *t* = 0 h and *t* = 48 h are shown on the right. Wound area is highlighted in light gray. **g–i**, Western blot analysis (**g**, **h**) and quantification of phospho-Akt (Ser473) (**i**) in primary dermal LECs from *PIK3CA^H1047R;Vegfr3-CreER^T2* mice treated in vitro with 4-OHT and/or VEGF-C for indicated times. Data are representative of one experiment (**g**) or three independent experiments (**h**). Data in (**i**) represent mean (*n* = 3 independent experiments) ± s.e.m. Two-tailed one-sample *t*-test (untreated *Ctrl* vs. *PIK3CA* in (**i**)) or Two-tailed unpaired Student's *t*-test (all others). Scale bars: 50 μm (**b–e**), 300 μm (**f**). Source data are provided as a Source Data file.

downstream signaling in LECs. 4-OHT-induced expression of the constitutive active p110α in primary LECs isolated from the *PIK3CA^H1047R;Vegfr3-CreER^T2* mice led to increased Akt activation under basal growth conditions in both sparse and confluent cells (Fig. 3g). To assess the response to growth factor activation, serum starved LECs were stimulated with VEGF-C, the major regulator of LEC migration[31]. As expected, serum starved 4-OHT treated *PIK3CA^H1047R*-expressing LECs showed again increased Akt phosphorylation compared with vehicle-treated cells in the absence of growth factor simulation (Fig. 3h, i). VEGF-C stimulation induced a comparable increase in Akt phosphorylation in control LECs. *PIK3CA^H1047R*-expressing LECs however showed a further increase in Akt activity in response to VEGF-C (Fig. 3h, i).

Taken together, VEGF-C stimulation induces hyperactivation of Akt in *PIK3CA^H1047R*-expressing LECs. This is in agreement with previous studies in cancer cells showing that the *H1047R* mutation residing in the kinase domain of p110α relies on upstream receptor tyrosine kinase (RTK) stimulation to reach a critical threshold of membrane binding and activity needed for transformation[32].

### *PIK3CA^H1047R*-driven LM is dependent on VEGF-C signaling.
To investigate the potential involvement of VEGF-C signaling in LM pathology in vivo, we analyzed expression of the pathway components in mouse LM lesions in comparison to normal vasculature. The VEGF-C receptors VEGFR3 and NRP2, but not VEGFR2, were strongly upregulated in the abnormal sprouts in the *PIK3CA^H1047R;Vegfr3-CreER^T2* ears (Fig. 4a, b, Supplementary Fig. 6a). Increase in VEGFR3, which correlated with increase in lymphatic vessel branching, was also observed in a knock-in mouse model[33], where *Pik3ca^H1047R* expression is driven by the endogenous promoter after Cre recombination (Supplementary Fig. 6a–c), as well as in the localized lesions in the *PIK3CA^H1047R;Prox1-CreER^T2* mice (Supplementary Fig. 6d). We could not, however, detect a significant increase in VEGFR3 mRNA or protein levels in primary *PIK3CA^H1047R*-expressing LECs (Supplementary Fig. 6e, f), suggesting that upregulation observed in vivo is not due to direct regulation by PI3K. As seen in human patients, we observed an accumulation of immune cells around advanced lesions in the *PIK3CA^H1047R;Prox1-CreER^T2* mice (Fig. 4c). In agreement with this, analysis of ears in the model of progressive microcystic LM showed a modest increase in the number of macrophages (Fig. 4d), that are the major source of VEGF-C[34,35]. These results suggest increased VEGF-C signaling

in LM, which may critically contribute to driving pathological vessel growth.

Next, we analyzed the response of *PIK3CA*-driven vascular malformations to inhibition of VEGF-C signaling in vivo. For this purpose, we used the model of progressive microcystic LM in postnatal mice (Supplementary Fig. 4e). To achieve continuous inhibition of the VEGF-C pathway, we used adeno-associated vectors (AAVs) encoding the soluble VEGF-C-trap (AAV-VEGFR3[1-4]-Ig[36]). Control mice were treated either with PBS or AAVs encoding the non-ligand binding region of the VEGFR3 extracellular domain (AAV-VEGFR3[4-7]-Ig). Soluble VEGFR3$_{1-4}$-Ig protein was detected in the serum 3 weeks after intraperitoneal AAV administration at a concentration of 121 ± 30 ng μl$^{-1}$ (*n* = 5) (Supplementary Fig. 7a). In agreement with previous studies[37], this was sufficient to block VEGF-C-induced lymphangiogenesis in the ear skin (Supplementary Fig. 7b).

Administration of AAVs to *PIK3CA^H1047R;Vegfr3-CreER^T2* mice one week after induction of lymphatic overgrowth (Supplementary Fig. 4e, Fig. 4e) led to an efficient inhibition of the growth of LM (Fig. 4f, g). As previously reported[38], VEGF-C inhibition did not affect mature vasculature in control mice (Fig. 4f). VEGF-C inhibition specifically affected the growth of lymphatic lesions, as it did not have an effect on the growth of lesions within the blood vasculature in *PIK3CA^H1047R;Cdh5-CreER^T2* mice that express the constitutive active p110α in all ECs (Supplementary Fig. 8a–d).

### Co-inhibition of VEGF-C and mTOR promotes LM regression.
Rapamycin (also known as sirolimus) is an allosteric inhibitor of the PI3K downstream target mTOR that has been used in the treatment of patients with vascular malformations including both VM and LM[3]. To avoid body weight loss associated with treatment with a high dose of Rapamycin[39], mice were administered with daily intraperitoneal injections of 10 mg kg$^{-1}$ of Rapamycin during 5 consecutive days. As expected, Rapamycin inhibited the overgrowth of both blood and lymphatic vessels in the *PIK3CA^H1047R;Cdh5-CreER^T2* mice when administered at the time of 4-OHT-mediated induction of *PIK3CA^H1047R* expression (Supplementary Fig. 9a–c). In addition, it significantly reduced vessel growth and hyperbranching when administered one week after induction of vascular overgrowth (Supplementary Fig. 9d–f).

To test the therapeutic effect on more advanced lesions, we initiated Rapamycin treatment two weeks after induction of vascular overgrowth in the *PIK3CA^H1047R;Vegfr3-CreER^T2* model

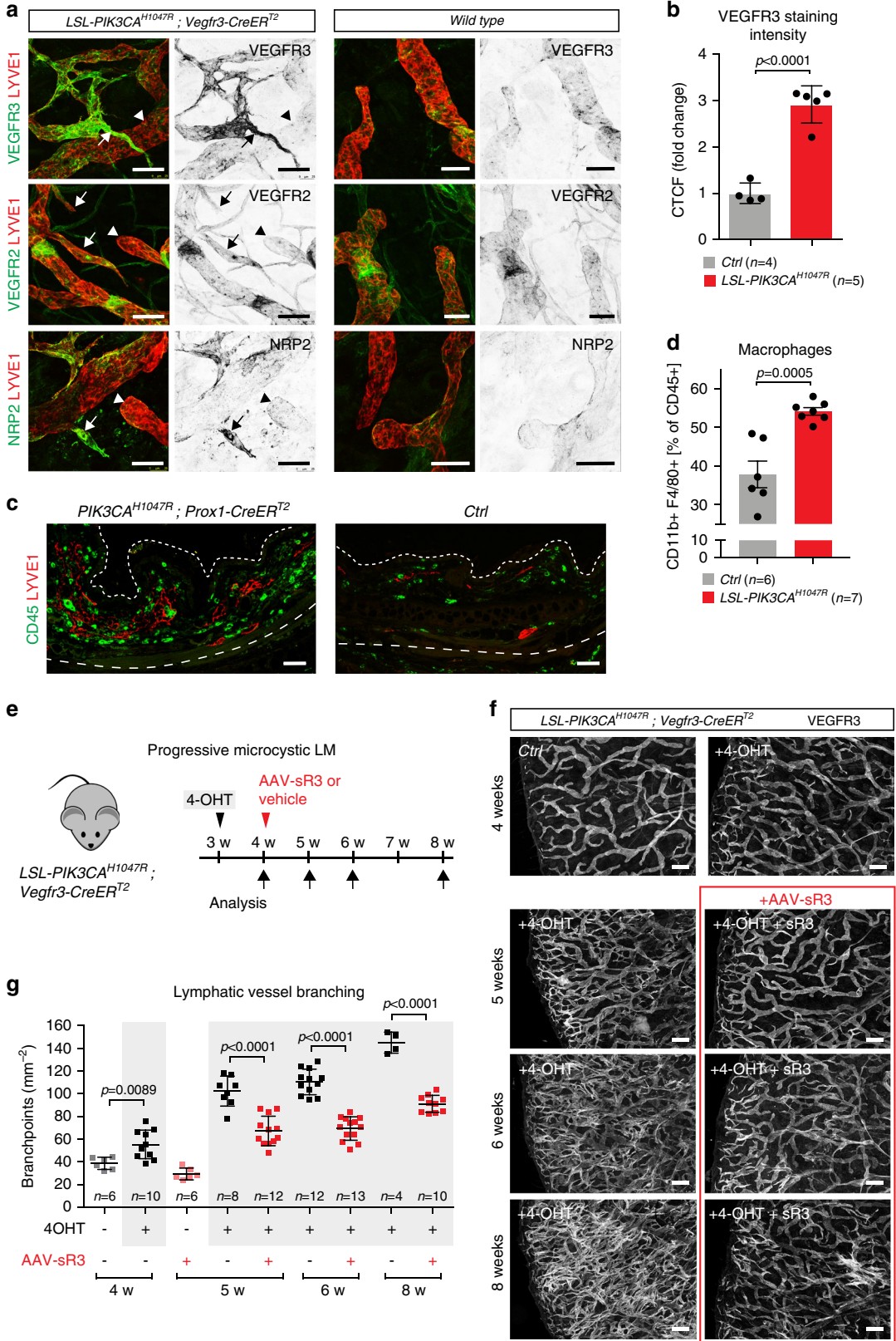

of progressive microcystic LM (Fig. 5a). Analysis of the vasculature after 1.5 weeks of Rapamycin treatment, administered every second day, showed only a modest inhibition of vascular growth during the treatment period, and no regression (Fig. 5b, c). Treatment with the VEGF-C trap also inhibited the growth but did not promote regression of the lymphatic vasculature (Fig. 5b,

c). However, combined treatment with the VEGF-C trap and Rapamycin showed an additive effect and promoted regression of the vasculature (Fig. 5b, c), and effectively blocked LEC proliferation (Fig. 5d). Remaining areas of hyperbranched vasculature in mice treated with both the VEGF-C trap and Rapamycin frequently showed blunted morphology of vessel

**Fig. 4 PIK3CA^H1047R-driven LM is dependent on VEGF-C signaling. a** Whole-mount immunofluorescence of ear skin of 5 weeks old *PIK3CA^H1047R;Vegfr3-CreER^T2* (left panels) mice showing upregulation of VEGFR3 and NRP2 expression but unchanged VEGFR2 levels in abnormal sprouts (arrows) in comparison to capillaries with normal appearance (arrowheads), and those in wild-type mice (right panels). 4-OHT (100 μg) was administered at P21. **b** Quantification of VEGFR3 staining intensity in the dermal lymphatic vasculature in the ear skin of 4-OHT-treated *PIK3CA^H1047R;Vegfr3-CreER^T2* mice ($n = 5$), compared with littermate controls ($n = 4$) ± s.d. **c** Immunofluorescence staining of paraffin sections of ear skin showing increase in CD45^+ immune cells around LYVE1^+ lymphatic lesions in a *PIK3CA^H1047R;Prox1-CreER^T2* mouse compared with a control (*Ctrl*). **d** Percentage of CD45^+CD11b^+F4/80^+ macrophages relative to all CD45^+ cells in the ear skin of 5 weeks old *PIK3CA^H1047R;Vegfr3-CreER^T2* mice treated with the vehicle ($n = 6$) or 4-OHT (100 μg, *PIK3CA^H1047R*; $n = 7$) at P21 (mean ± s.d.). **e** Experimental plan for the induction of progressive microcystic LM and inhibition using the soluble VEGF-C trap (AAV-VEGFR3-Ig; AAV-sR3) or vehicle (PBS). **f** Whole-mount staining of ears from untreated *PIK3CA^H1047R;Vegfr3-CreER^T2* mice (*Ctrl*), and mice treated with 4-OHT (100 μg) and AAV-sR3 or vehicle, and analyzed at different stages after induction. Images in red frame show immunofluorescence of ears from AAV-sR3 treated mice. **g** Quantification of lymphatic vessel branching in the progressive LM model. Red squares represent data from AAV-sR3 treated mice. Data represent mean ($n$ = number of ears as indicated) ± s.d. Two-tailed unpaired Student's *t*-test. Scale bars: 50 μm (**a**), 200 μm (**c**, **f**). Source data are provided as a Source Data file.

sprouts, compared with spiky appearance characteristic of active sprouting process in vehicle-treated mice (Fig. 5e, f).

## Discussion

LM are vascular anomalies presenting with large fluid-filled cysts (macrocystic) or lesions that infiltrate tissues (microcystic) and cause morbidity, pain and organ dysfunction. Using a mouse model of LM driven by an activating *PIK3CA* mutation, we here characterize molecular and cellular mechanisms of lymphatic lesion formation and therapy response. Notably, we show that the progressive growth of *PIK3CA^H1047R*-driven microcystic LM is dependent on the upstream lymphangiogenic VEGF-C/VEGFR3 signaling, which permits a novel effective therapeutic intervention to treat LM.

Here we focused on the common activating *PIK3CA* mutation H1047R causative of both VM and LM[5–7], but also cancer and overgrowth syndromes[9]. We found that the H1047R mutation is causative of both macrocystic and microcystic LM in a LEC-autonomous manner, with the developmental timing of *PIK3CA* activation determining the LM subtype. In mouse skin, embryonic LEC-specific activation of *PIK3CA* led to macrocystic lesions characterized by large cysts localized predominantly to the neck region of the skin, similarly as in human LM patients. In contrast, late embryonic or early postnatal activation caused microcystic malformations with hyperbranched lymphatic lesions. Why activation of the same signaling pathway can cause different LEC responses leading to morphologically different types of malformations remains to be studied. It is possible that the cellular response may differ depending on the developmental origin and differentiation status[40] (progenitor vs. lining a lumenized vessel) as well as the cellular state (proliferating vs. quiescent). Differences in the tissue environment may also play a role. For example, tissue stiffness and composition of the immune cells that are both known to regulate LEC responses and lymphangiogenesis[34,41,42] differ between embryonic and postnatal skin.

The development of several types of vascular malformations, including cerebral cavernous malformations (CCM) and arteriovenous malformations (AVM) require active angiogenesis. The formation of these vascular lesions can be efficiently induced experimentally in mice during developmental vascular growth, but require reactivation of quiescent endothelium at adult stages[43–47]. The congenital nature of LM in human suggests that lymphatic endothelium is similarly sensitive to activation of p110α signaling during the active growth of the vasculature, yet lesions may also appear later in life and show progressive growth. Using the mouse model of *PIK3CA^H1047R*-driven LM, we found that, as expected, both embryonic and early postnatal lymphatic vasculature is responsive to p110α activation, leading to formation of localized vascular lesions. However, LEC-specific activation of *PIK3CA^H1047R* expression at 3 weeks of age led to a

generalized, rather than localized response characterized by progressive lymphatic vascular hyperplasia. This is in agreement with a previous study modeling generalized lymphatic anomaly (GLA) in mice upon LEC-specific induction of *PIK3CA^H1047R* at 4 weeks of age[16]. The exquisite sensitivity of quiescent lymphatic endothelium to p110α activation may be related to the requirement of inhibition of the pathway for the maintenance of endothelial quiescence[48]. This finding raises the question of why *PIK3CA*-driven lymphatic vascular overgrowth is not a frequent manifestation in adult vasculature. It is possible that a critical threshold of mutant cells needed to cause vascular overgrowth is efficiently induced in our experimental model, but is not achieved in quiescent endothelium hit by a single somatic mutation. It is also possible that endothelial cells that acquire *PIK3CA* mutation require a permissive local environment to engage in signaling with normal neighbors that allow them to collectively drive pathological vascular growth. Clonal co-operation in vascular malformations, a concept that is well-established in cancer[49], may be supported by the observation that only a small proportion of endothelial cells forming CCM lesions in human carry a mutation[50,51]. In addition, in a mouse model of CCM, initial clonal expansion of *Ccm3* deficient cells was followed by incorporation and phenotypic change of normal cells driving further growth of the lesion[52,53]. Of interest, heterogeneity between neighboring LECs in the level of VEGFR3, which is regulated by PI3K/mTOR signaling (our study and[39,54]), was shown to non-cell-autonomously drive pathological vessel hyperplasia through cell contact-dependent regulation of Notch signaling[55]. Longitudinal magnetic resonance imaging of the LM lesions could provide important insights into the mechanisms of LM growth.

Macrocystic LM can be generally effectively treated with sclerotherapy or surgical resection. The treatment of microcystic LM is more challenging due to the infiltrative growth of the lesions. Ongoing clinical trials assessing the efficacy of Sirolimus (mTOR inhibitor Rapamycin) for the treatment of *PIK3CA*-driven vascular malformations have shown promising results. Sirolimus efficiently stops the growth of vascular malformations in mice and human, and improves the patients' quality of life[3,5,15–19]. However, regression of the lesion is achieved only in rare cases[3]. It is also currently unclear if patients need life-long treatment, which is associated with a risk of side effects of the drug. Consistent with the clinical data and previous studies on a mouse model of *PIK3CA*-driven generalized lymphatic anomaly[16], we found that Rapamycin inhibited lymphatic vascular growth in the experimental model of *PIK3CA^H1047R*-driven microcystic LM. The treatment resulted in an efficient inhibition of the initial growth and expansion of early lesions, but only a modest inhibition, and no regression, could be observed in more advanced lesions. Interestingly, in a mouse model of VEGF-C induced pulmonary lymphangiectasia, which persisted even after

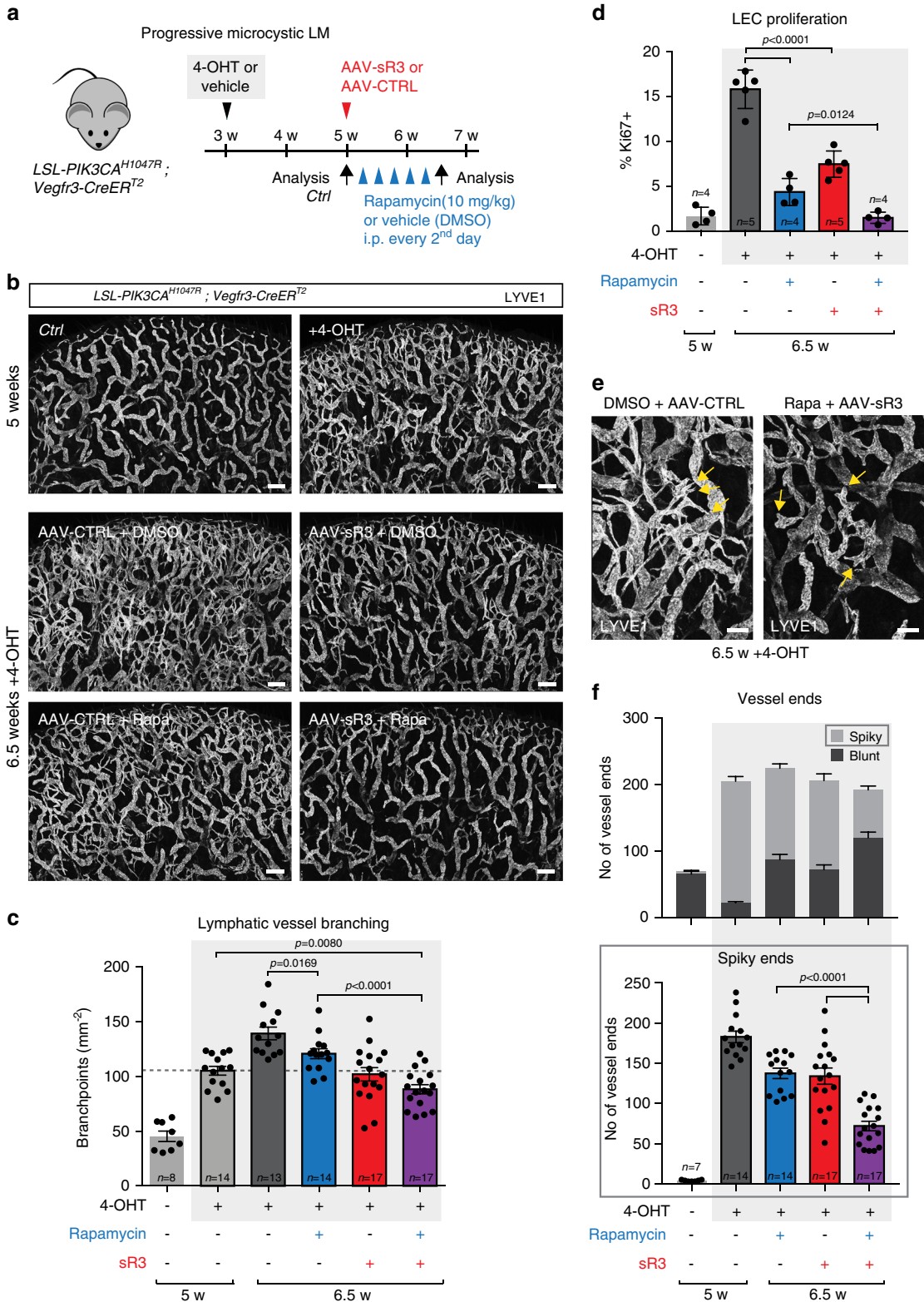

**a** Progressive microcystic LM

LSL-PIK3CA^H1047R ; Vegfr3-CreER^T2

**b** LSL-PIK3CA^H1047R ; Vegfr3-CreER^T2 — LYVE1

**c** Lymphatic vessel branching

**d** LEC proliferation

**e** DMSO + AAV-CTRL / Rapa + AAV-sR3 — 6.5 w +4-OHT

**f** Vessel ends / Spiky ends

normalization of VEGF-C levels, Rapamycin treatment effectively promoted regression of abnormal lymphatic vessels without affecting normal vessels[39]. However, a significant reversal of lymphangiectasia required a high dose (20 mg kg$^{-1}$ daily) of Rapamycin that was associated with body weight loss[39]. Although compatible with survival of adult mice[39], high dose of Rapamycin could not be tolerated during neonatal period of growth. It may thus not be feasible to use a high enough dose of Rapamycin

required for induction of lymphatic vessel regression in children and young adolescents. Interestingly, Alpelisib (BYL719), a specific PIK3CA inhibitor, has shown a higher efficacy than Rapamycin in the treatment of PIK3CA-related overgrowth syndromes (PROS) in mice and humans[56]. Alpelisib improved organ abnormalities and dysfunction, also in the vasculature, with minimal side effects. Future studies should address the efficacy of Alpelisib in patients with LM.

**Fig. 5 Co-inhibition of the upstream VEGF-C/VEGFR3 and the downstream mTOR signaling pathways promotes LM regression. a** Experimental plan for the induction and treatment of progressive microcystic LM using a combination therapy with VEGF-C trap and Rapamycin. **b** Whole-mount staining of ears from *PIK3CA^H1047R;Vegfr3-CreER^T2* control (*Ctrl*) and 4-OHT-treated (100 μg) mice at 5 weeks of age, or following a 1.5-week treatment with Rapamycin, AAV-sR3 and/or vehicles at 6.5 weeks of age. **c** Quantification of lymphatic vessel branching in the *PIK3CA^H1047R;Vegfr3-CreER^T2* mice. Stages of analysis and treatments are indicated. Data represent mean (*n* = number of ears as indicated) ± s.e.m. **d** Flow cytometry analysis of dermal LEC proliferation in the *PIK3CA^H1047R;Vegfr3-CreER^T2* mice treated as indicated. Data represent mean (*n* = number of mice as indicated) ± s.d. **e** Lymphatic vessel morphology in areas of hypersprouting in 6.5-week-old 4-OHT-treated *PIK3CA^H1047R;Vegfr3-CreER^T2* ears. Note blunt morphology of lymphatic sprouts in mice treated with Rapamycin and AAV-sR3 compared with vehicle-treated mice (yellow arrows). **f** Quantification of all vessel ends and their morphology (upper graph), with data shown separately for spiky ends representing active sprouts (lower graph), in the indicated groups. Data represent mean (*n* = number of mice as indicated ± s.e.m.) Two-tailed unpaired Student's *t*-test. Scale bars: 200 μm (**b**), 100 μm (**e**). Source data are provided as a Source Data file.

In addition to the cell-autonomous migration-promoting effect of p110α on LECs observed in vitro, lymphangiogenic VEGF-C/VEGFR3 signaling in the LM lesion was found to significantly contribute to the further vascular growth of the lesions in vivo. Strong upregulation of the VEGF-C receptors VEGFR3 and NRP2 was observed in the lymphangiogenic sprouts, consistent with previous observations in LM patients[57,58]. In addition, we observed increased infiltration of immune cells, including macrophages, which are an important source of pro-lymphangiogenic growth factors such as VEGF-C[34,35]. Although LECs expressing the oncogenic p110α showed increased Akt activity already under basal conditions, VEGF-C stimulation led to hyperactivation of the pathway. These findings suggest a positive feedback reinforcing the upstream pathway that subsequently promotes abnormally high activity of the downstream signaling, making the pathway a potential target for therapeutic intervention. Indeed, we found that inhibition of VEGF-C/VEGFR3 signaling using the soluble VEGF-C trap inhibited LM lesion growth in vivo. Notably, treatment with the soluble VEGF-C trap was more effective than Rapamycin in inhibiting LM growth, and when administered in combination with Rapamycin it was able to promote lymphatic vessel regression. Our results suggest that, like in cancer, optimal therapeutic benefit may require targeting both the endothelial cell-autonomous *PIK3CA*-driven signaling and the microenvironment-derived paracrine signaling that contribute to driving pathological vessel growth. The relative contributions of the two mechanisms may be different at different stages of lesion formation, which can affect response to therapy. It is possible that the beneficial effect of Rapamycin is partly due to its immuno-suppressive functions, although this should be addressed in future studies.

In conclusion, we show that the progressive growth of *PIK3CA^H1047R*-driven microcystic LM is dependent on the upstream lymphangiogenic VEGF-C/VEGFR3 signaling. Combined inhibition of VEGF-C signaling using the soluble VEGF-C trap and the PI3K downstream target mTOR using Rapamycin, but neither treatment alone, promotes the regression of experimental LM in mice. These results suggest that the best therapeutic outcome for *PIK3CA^H1047R*-driven microcystic LM is achieved by co-inhibition of the VEGF-C/VEGFR3 and the downstream (mTOR) pathways. Therapies targeting the key upstream pathway in combination with PI3K pathway inhibition may be relevant also for other *PIK3CA*-driven pathologies.

## Methods

**Patients.** Residual tissue of macrocystic and microcystic LMs were collected from patients undergoing therapeutical surgery. All participants gave their informed consent, as approved by the ethical committee of the Medical Faculty of the University of Louvain, Brussels, Belgium. All protocols were compliant with the Belgian laws governing research on human subjects. The phenotypes of the patients were evaluated as accurately as possible by the referring clinicians. DNAs were extracted from crunched (powderized) snap-frozen tissues, using Wizard genomic DNA purification kit (Promega). DNAs were screened by Ion Torrent technology with a custom Ampliseq panel (www.ampliseq.com) designed to cover the coding exons of *PIK3CA*. Genetic variants were analyzed using Highlander software (http://sites.uclouvain.be/highlander/). Histology was examined by a pathologist expert in vascular anomalies.

**Mouse lines and treatments.** *R26-LSL-PIK3CA^H1047R* [22], knock-in *LSL-Pik3ca^H1047R* [33], *Cdh5-CreER^T2* [59], *Vegfr3-CreER^T2* [23], *Prox1-CreER^T2* [24], and *R26-mTmG* [60] mice were analyzed on a C57BL/6J background. Both female and male mice were used for analyses and no differences in the phenotype between the genders were observed. The morning of vaginal plug detection was considered as embryonic day 0 (E0). For inducing recombination during embryonic development 4-hydroxytamoxifen (4-OHT, H7904, Sigma) dissolved in peanut oil (10 mg ml⁻¹) was administered intraperitoneally to pregnant females. For postnatal induction, 4-OHT dissolved in acetone (10 mg ml⁻¹), was applied topically to the dorsal side of the right ear of each mouse. The 4-OHT doses and days of administration are indicated in the figures and/or figure legends and summarized in Supplementary Table 1. Additional information on the routes of administration and the effect are provided in Supplementary Table 1. For inhibition of VEGF-C signaling, we used adeno-associated viral (AAV)-derived VEGF-C trap encoding the ligand binding domains 1–4 of VEGFR3, fused to the IgG Fc domain (AAV9-mVEGFR3₁₋₄-Ig) and control AAVs that encoded the inactive domains four to seven of VEGFR3-Ig (AAV9-mVEGFR3₄₋₇-Ig)[36]. Mice received a single intraperitoneal injection of $1 \times 10^{10}$ virus particles diluted in PBS at a total volume of 100 μl at the indicated time point. PBS or AAV9-mVEGFR3₄₋₇-Ig were used as control treatments. Rapamycin (R-5000 Rapamycin, >99% LC Labs) was dissolved in DMSO at 10 mg ml⁻¹ and injected intraperitoneally either once per day (Supplementary Fig. 9) or every second day (Fig. 5) at the dose of 10 mg kg⁻¹. DMSO was used as a control. A single dose ($2.5 \times 10^{10}$ viral particles per ear in 10 μl) of AAVs encoding VEGF-C[61,62] was injected intradermally into both ears of 5 weeks old adult mice. All experimental procedures were approved by the Uppsala Animal Experiment Ethics Board and performed in compliance with all relevant Swedish regulations, or the Catalan Departament d' Agricultura, Ramaderia i Pesca, following protocols approved by the local Ethics Committees of IDIBELL-CEEA.

**Antibodies.** The details of primary antibodies used for immunofluorescence of whole mount tissues, cells and paraffin-embedded sections are provided in Supplementary Table 2. Secondary antibodies conjugated to Cy3, Alexa Fluor 405, 488 or 647 were obtained from Jackson ImmunoResearch.

**Immunofluorescence.** Whole-mount tissue was fixed in 4% paraformaldehyde at room temperature for 2 h followed by permeabilization in 0.3% Triton X-100 in PBS (PBST) for 10 min and blocking in PBST plus 3% milk for 2 h. Primary antibodies were incubated at 4 °C overnight in blocking buffer and washed in PBST before incubating them with fluorescence-conjugated secondary antibodies in blocking buffer for 2 h at room temperature. Stained samples were then washed and mounted in Mowiol. Cells were fixed with 4% paraformaldehyde at room temperature for 20 min, washed twice with PBS and permeabilized in 0.5% Triton X-100 in PBS for 5 min, and blocked in PBST plus 2% BSA for 1 h. Cells were incubated with primary antibodies diluted in the blocking buffer at room temperature for 1 h. After washing three times with PBS plus 0.2% BSA, the cells were incubated with fluorescence-conjugated secondary antibodies at room temperature for 45 min, followed by staining with DAPI (D9542, Sigma, 1:1000) for 5 min, and fixing with 4% paraformaldehyde for 4 min before further washing and mounting in Mowiol.

**Immunostaining of paraffin sections.** Tissue was fixed in 10% neutral buffered formalin (human) or 4% paraformaldehyde (mouse) and embedded in paraffin. Deparaffinization and rehydration steps were performed manually on 4 μm (human) or 5–6 μm (mouse) paraffin sections. Heat-induced antigen retrieval was performed using Sodium Citrate buffer (10 mM, pH 6) at 95 °C. Immunohistochemistry with antibodies against PROX1 and PDPN (Supplementary Table 1) were performed on an automatic immunostainer, Ventana Benchmark Ultra. For immunofluorescence, paraffin sections were blocked with TNB (Tris-NaCl-blocking buffer, prepared according to the Tyramide Signal Amplification kit (TSA™, NEN™ Life Science Products) and washed with TNT buffer (0.1 M Tris pH 7.5,

0.15 M NaCl, 0.05% Tween 20). The TSA™ kit was used to enhance and detect the signal of goat anti-mouse VEGFR3 (AF743, R&D Systems) and rat anti-mouse CD45 antibody (ab25386, Abcam) in the mouse paraffin sections and mouse anti-human VEGFR3 (clone 9D9F9, Millipore) antibody in the human paraffin sections. All other primary antibodies (see Supplementary Table 1) were used at 4 °C overnight and visualized with fluorescence-conjugated secondary antibodies after the TSA kit. The majority of the slides were mounted in Mowiol except the ones with CD45 primary antibody, where Autofluorescence Quenching kit (Vector® TrueVIEW™) together with VECTASHIELD® HardSet™ Antifade Mounting Media was used to decrease autofluorescence.

**Primary LEC culture.** Murine primary dermal LECs were isolated from the tail skin of 6-13 weeks old *LSL-PIK3CA*[H1047R];*Vegfr3-CreER*[T2] mice by sequential selection with PECAM1 (Mec13.3, Pharmingen) and LYVE1 (Aly7, Abnova) antibodies bound to Dynabeads, as described previously[23]. Mouse LECs were cultured on 0.5% gelatin (G1393, Sigma)-coated dishes in complete DMEM medium. To induce Cre-mediated recombination, 4-OHT was dissolved in absolute ethanol and added to the culture medium at the final concentration of 5 μM for 48 h. Cell were then reseeded in 12-wells at (low) density of $3 \times 10^4$ cells per cm² for western blot analysis and qRT-PCR analysis. For serum starvation, cells were reseeded after 4-OHT treatment at the low density and starved overnight in 0.5% fetal bovine serum (FBS) containing medium and stimulated with VEGF-C (R&D Systems, 200 ng ml⁻¹) for time periods indicated in the figures. To assess cell migration in vitro, wound-healing assays were performed with IncuCyte ZOOM™ according to the manufacturer's instructions. Cells were reseeded in 96-well ImageLock plate (Essen BioScience) at density of $6.25 \times 10^4$ cells per cm² after 4-OHT treatment. WoundMaker™ (Essen BioScience) was used to create homogeneous scratch wounds, and the cells were then washed twice with fresh medium to remove any debris, followed by adding 100 μl fresh complete DMEM medium supplemented with or without VEGF-C (25 ng ml⁻¹) to each well. The plate was placed into the IncuCyte ZOOM™ apparatus and images of cells were acquired automatically every 1 h for a total duration of 72 h. All cells were cultured at 37 °C in a humidified atmosphere with 5% $CO_2$.

**Western blot analysis.** Total protein extract was obtained by lysing cells in lysis buffer [150 mM NaCl, 1% Triton X-100, 0.5% Sodium deoxycholate, 0.1% SDS, 50 mM Tris-HCl (pH 7.6) supplemented with 25 mM NaF, 1 mM $Na_3VO_4$ and protein inhibitor cocktail (11873580001, Roche)]. Cell lysates were spinned for 10 min at 13200 r.p.m. at 4 °C and supernatants were collected. Protein concentration was determined using a BCA™ Protein Assay Kit (23227, Thermo Fisher Scientific) according to manufacturer's instructions and measured using Gen5 All-In-One microplate reader. Equal amounts of proteins were loaded and separated in 4-20% gradient polyacrylamide gels (Invitrogen), transferred to a polyvinylidene difluoride membrane (PVDF; 88520, Thermo Fisher Scientific) and blocked for 1 h at room temperature in 1× tris buffered saline tween (TBST) [150 mM NaCl, 10 mM Tris-HCl (pH 7.4) and 0.1% Tween] containing 5% (w/v) bovine serum albumin (BSA). The membranes were subjected to overnight incubation at 4 °C with primary antibodies diluted in 1× TBST plus 5% BSA. Following this incubation, membranes were rinsed three times with 1× TBST for 5 min each and incubated for 1 h at room temperature with HRP-conjugated secondary antibodies (diluted in 1× TBST plus 5% BSA). Membranes were rinsed three times with TBST for 5 min each and specific binding was detected using the enhanced chemiluminescence (ECL) system (WP20005, Invitrogen) and the ChemiDoc XRS + System (Biorad). All images were collected using Image Lab software. Protein molecular masses were estimated relatively to the electrophoretic mobility of co-transferred prestained protein marker (26634, Thermo Fisher Scientific). The following antibodies were used: rabbit anti-mouse phospho-Akt (Ser473) (#4060), phospho-Akt (Thr308) (#2965), Akt (#9272), VEGFR2 (#2479), GAPDH (#2118, all from CST and 1:2000), goat anti-mouse VEGFR3 (AF743, R&D Systems, 1:1000).

VEGFR3$_{1-4}$-Ig and VEGFR3$_{4-7}$-Ig proteins in serum were detected by western blotting of 0.5 μl serum samples collected at the indicated time points as described[63]. 50, 100, and 200 ng of recombinant mouse VEGFR3 chimera protein (743-R3-100, R&D Systems) were used as a standard for calculating the concentration of VEGFR3$_{1-4}$-Ig and VEGFR3$_{4-7}$-Ig proteins. Mouse VEGFR3 domains 1–4 and 4–7 were detected with polyclonal goat anti–mouse VEGFR3 antibody (AF743, R&D Systems, 1:1000) against the extracellular domain of VEGFR3. The blots were quantified using ImageJ software. Full blots are shown in source data.

**qRT-PCR analysis.** Total RNA was isolated from dermal mouse LECs using RNeasy Mini kit (Qiagen). Complementary DNA (cDNA) was synthesized using Superscript VILO Master Mix (Invitrogen). cDNA was analyzed by real-time quantitative PCR (qRT-PCR) (StepOne Plus system, Applied Biosystems) using TaqMan gene expression assays and TaqMan gene expression Master Mix (Applied Biosystems). Relative quantifications of gene expression were performed using the comparative cycle threshold method (ΔCT) with *Gapdh* as the reference gene. TaqMan Assays used for mouse LECs were as followed: *Vegfr2* (Mm01222421_m1), *Vegfr3* (Mm01292604_m1), *Gapdh* (Mm99999915_g1). The values represent average relative gene expression.

**Flow cytometry.** For FACS analysis of proliferating cells ear skin of adult mice were dissected, cut into small pieces and digested in Collagenase IV (Life Technologies) 10 mg ml⁻¹, DNase I (Roche) 0.1 mg ml⁻¹ and FBS 0.5 % (Life Technologies) in PBS at 37 °C for about 30 min with vigorous shaking every 5 min. Collagenase activity was quenched by dilution with FACS buffer (PBS, 0.5% FBS, 2 mM EDTA) and digestion products were filtered twice through 70 μm nylon filters (BD Biosciences). Cells were again washed with FACS buffer and immediately processed for immunostaining first by blocking Fc receptor binding with rat anti-mouse CD16/CD32 (eBiosciences) followed by incubation with antibodies targeting PDPN, CD31/PECAM1, CD45 and CD11b (all obtained from eBioscience, see Supplementary Table 2). Two different panels of antibodies were used depending on if samples were taken from mice with *R26-mTmG* reporter or not. After staining, cells were washed with PBS and then stained for dead cells using the blue LIVE/DEAD® fixable dead cell stain kit (Life Technologies), followed by fixation and permeabilization using the Foxp3/Transcription factor staining kit (eBioscience) according to the manufacturer's instructions. Finally cells were incubated with rat serum and KI67 antibody (eBioscience). Cells were analyzed on a BD LSR Fortessa cell analyzer equipped with 5 lasers (355, 405, 488, 561, and 643 nm). Compensation was performed using the anti-rat/hamster compensation bead kit and the ArC amine reactive compensation bead kit (Life Technologies). Single viable cells were gated from FSC-A/SSC-A, FSC-H/FSC-W and SSC-H/SSC-W plots followed by exclusion of dead cells in the UV dump channel. FMO controls were used to set up the subsequent gating scheme to obtain cell populations and quantification of proliferating cells (Supplementary Fig. 10).

For FACS analysis of immune cells the ear skin was dissected into dorsal and ventral parts, cut into pieces and digested in Liberase TL (Roche) 100 μg ml⁻¹, DNase I (Roche) 0.5 mg ml⁻¹ in PBS with 0.2% FBS at 37 °C for 1.5 h at 500 r.p. m. Liberase TL activity was quenched by adding 2 mM EDTA and the product was filtered through 50 μm filters (CellTricks). The cells were washed with FACS buffer (PBS, 0.5% FBS, 2 mM EDTA) and incubated first with rat anti-mouse CD16/32 antibody (eBioscience) for blocking of the Fc receptor binding and then with CD45, CD11b and F4/80 targeting antibodies (Supplementary Table 2). Cell death was analyzed by adding SYTOX Blue dead stain (Life Technologies). The cell suspension was filtered again immediately before analysis. The analysis was obtained on CytoFLEX Flow Cytometer (Beckman Coulter). Single cells were gated from SSC-H/FSC-H and FSC-Width/FSC-H plots followed by gating for CD45⁺ viable cells and CD11b⁺F4/80⁺ cells (Supplementary Fig. 9). All flow data was processed using FlowJo software version 10.5.0 (TreeStar).

**Image acquisition and quantification.** All confocal images represent maximum intensity projections of Z-stacks of single tiles or multiple tile scan images. Images were acquired using Leica SP8 confocal microscope or Leica DMi8 fluorescence microscope with Leica LAS X software. Quantification of lymphatic vessel branching was done with Angiotool plugin of ImageJ (version 2.0.0-rc-65/1.5lu) using tile scan images of adult ear skin (xy = 3260 μm × 2200 μm, tip region) or dorsal skin of E17 embryos (xy = 2322 μm × 4644 μm, cervical/thoracic region). Quantification of VEGFR3 staining intensity was done by measuring corrected total cell fluorescence (CTCF) = integrated density – (area of selected cell × mean fluorescence of background readings) using ImageJ, and the vessel area was determined by combined staining of LYVE1 and VEGFR3. Vessel ends with blunt or spiky morphology were marked with Photoshop CS6 software and counted manually using tile scan images of adult ear skin (xy = 3260 μm × 2200 μm at the tip of the ear excluding the region 34 μm from the edge of the ear) and presented as number of ends per ear. All quantifications include littermate controls; for *Cdh5-CreER*[T2] model Cre⁻ littermates treated with 4-OHT were used (Supplementary Figs. 8, 9), *Vegfr3-CreER*[T2] model Cre⁺ untreated (Figs. 4, 5) or vehicle-treated (Figs. 4, 5) mice were used. At least two different litters of mice were analyzed. For validation of the purity of isolated primary LEC populations, % of VE-cadherin⁺ cells and % of PROX1⁺ cells were quantified using maximum intensity projection images based on staining of DAPI, VE-cadherin and PROX1 (n = 1386 cells in total from 15 experimental conditions and 2 batches of cells). The cells were marked using Photoshop CS6 software and counted manually.

The images acquired by IncuCyte ZOOM™ were processed and analyzed using IncuCyte™ scratch wound cell migration software module. Cell density in the wound area expressed relative to the cell density outside of the wound area over time (relative wound density%) was quantified. The quantification of images from western blots was done using ImageJ software.

**Statistics and reproducibility.** Graphpad Prism was used for graphic representation and statistical analysis of the data. Data between two groups were compared with One-sample *t*-test (Fig. 3i), or paired (Supplementary Fig. 4d) or unpaired (all others) two-tailed Student's *t*-test, assuming equal variance. Differences were considered statistically significant when $P < 0.05$. The experiments were not randomized, and no blinding was done in the analysis and quantifications. No statistical methods were used to predetermine the sample size. All microscopy images of mouse tissues are representative of a minimum n = 3 mice except for Fig. 3c, wild-type embryonic back skin n = 1.

**Reporting summary**. Further information on research design is available in the Nature Research Reporting Summary linked to this article.

## Data availability

The next-generation sequencing data that support the findings of this study are available on request from MV. The data are not publicly available due to them containing information that could compromise research participant consent. All other data supporting the findings of this study are available from the corresponding author (TM) upon request. The source data underlying Figs. 2c, 3b, f–i, 4b, d, g, 5c, d, f and Supplementary Figs. 2e, 4d–h, 5a, b, 6b, c, e, f, 7a, b, 8c, d, 9c, f are provided as a Source Data file.

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

## Acknowledgements
We thank Sagrario Ortega (CNIO, Madrid) for the *Vegfr3-CreER^{T2}* mice and Ralf Adams (Max Planck Institute for Molecular Biomedicine, Münster) for the *Cdh5-CreER^{T2}* mice. We also thank the BioVis facility (Uppsala University, Sweden) for flow cytometer usage and support, and Simon Stritt for help with flow cytometry and Sofie Lunell-Sergerqvist for technical assistance. This work was supported by the Swedish Cancer Society (CAN 2016/535), the European Research Council (ERC-2014-CoG-646849), Knut and Alice Wallenberg Foundation (2015.0030) and the Swedish Research Council (542-2014-3535) to T.M.; the F.R.S.-FNRS (Fonds de la Recherche Scientifique, Belgium), grant [T.0026.14] to M.V. and [T.0146.16] to L.B., and the Walloon Excellence in Lifesciences & BIOtechnology (FNRS-WELBIO) grant [WELBIO-CR-2010-15R] and the Fund Generet managed by the King Baudouin Foundation to M.V.; and Academy of Finland (grants 314498, 320249) to K.A. Laboratories of M.V. and T.M. are part of V.A.Cure, and thus received funding from the European Union's Horizon 2020 research and innovation programme under the Marie Skłodowska-Curie grant agreement No 814316. P.B. is a Senior Platform Manager of the University of Louvain. MG lab is supported by Ministerio de Ciencia, Innovación y Universidades, which is part of Agencia Estatal de Investigación (AEI, Spain) through the projects SAF2017-89116-R co-funded by European Regional Developmental Fund (ERDF), a Way to Build Europe; by la Fundació Bancària "La Caixa"; by the CERCA program of la Generalitat de Catalunya. S.D.C. is a recipient of a fellowship from the European Union's Horizon 2020 Research and Innovation Programme under the Marie Sklodowska-Curie grant agreement No. 749731. Open access funding provided by Uppsala University.

## Author contributions
I.M.C., Y.Z., M.P., H.O., S.D.C., M.G., M.V., and T.M. designed experiments and analyzed data; I.M.C., Y.Z., M.P., H.O., S.S., and S.D.C. performed experiments; P.B., L.L., L.B., and M.V. provided clinical patient data and material; D.S. and K.A. provided essential tools and advice; T.M. supervised the project and wrote the manuscript with input from other authors. All authors discussed the results and commented on the manuscript.

## Competing interests
K.A. and T.M. have filed a patent application related to the manuscript as inventors. Other authors declare no competing interests.
