## [Peer Review File · Nature Communications]

Reviewers' Comments:

Reviewer #1:

Remarks to the Author:

The paper is well-done and convincing. It is important and has significant clinical translational potential. I believe it should be published in Nature Communications. I don't have any major edits for the manuscript. Minor suggestions include: 1) Figure 1 compares microcystic vs macrocystic lesions histologically but the title of the figure describes both lesions having the same mutation- I would add to the title the specific PIK3CA mutation both lesions have. Also, there is a typo in the legend that should say macrocystic and not microcystic. The second figure would be strengthened if they had clinical photographs of the lymphatic malformations in the mice as well as MRI images to further show they recapitulate the human disease. Lastly, they could add to the discussion that their finding that cervical LMs are more likely to be macrocystic mirrors the human conditions as well (this further strengthens their data).

Reviewer #2:

Remarks to the Author:

Martinez-Corral et al have investigated lymphatic malformations associated with macrocystic and microcystic lesions. They used human samples (n=5) and generated a mouse model with inducible LEC activation of the main causative mutation of PI3KCA (H104R). The activation of this mutated form has been induced in different mice (using different promoters), at embryonic (E11) and post-natal (P2-5) stages. These approaches led to cysts and lymphatic vessel hypersprouting. The timing of activation determines the type of malformation (macrocystic when induced at an early stage and microcystic at a later stage). The authors provide clear evidence that both microcystic and macrocystic lesions driven by the mutation are dependent to VEGF-C signaling. Importantly, both malformations can be reduced by an inhibitory combination of VEGF-C/VEGFR-3 and PI3K/mTOR pathways.

This is a timely study well conducted and convincing that is providing innovative data with therapeutic potential. There are no major comment.

Specific points:

- page 6: Stainings of PDPN, Prox1 and Lyve-1 should be illustrated (not as data not shown).
- Figure 3: Zipper junctions are illustrated as a hallmark of lymphangiogenic vessels. However, a comparison with WT tissue should be provided.
- Faster wound closer is detected under basal conditions. The dependence to VEGF-C should be documented or at least discussed.
- Does the mutation activation in primary LEC affect VEGFR-3 and VEGFR-2 levels and activation by VEGF-C? Do cells respond equally to WT VEGF-C and mutated VEGFC156S? This could be tested, at least in the scratch assay.

Reviewer #3:

Remarks to the Author:

In this article, Dr. Martinez Corral and coauthors explore the role played by PIK3CA mutation in lymphatic cells. Using different mouse models, they observed different subgroup of lymphatic malformation and identified a dual therapy as potential new therapeutic opportunity for patients. The paper is well written, easy to read but I have some important concerns regarding the interpretation.

1-The first conclusion is potentially wrong. Indeed, it is difficult to say that the occurrence of an early mutation during embryological development is leading to macrocystic lesion and late mutation to microcystic lymphatic malformation. In the first case the authors have injected 4 OH

tamoxifen to pregnant mice (inducing indeed a systemic recombination in deep lymphatic cells) and on the other hand they applied locally 4OHT which is inducing the mutation only in superficial skin lymphatic vessels. The message is indeed completely different and overstated. By the way, this technical part is not well described and I was confused when I was reading the results.

2-Several mandatory controls are missing: Fig. 3a (LYVE1 pS6 staining in normal lymphatic vessels at baseline...), Fig. 4A...

3-I am concerned about the WB. Why the authors do not detect any AKT phosphorylation in their primary cultured cells? It is very strange and unusual. Furthermore, the IF in Fig. 3A shows activation of mTORC1, a lower step in the cascade. IF of PAKT but also WB of PS6 are requested. Of note, this signaling pathway is extremely sensitive to starving (mandatory before to interpret any in vitro and in vivo data).

4- Fig. 4a: VEGFR2 staining seems to be nuclear. Again, no controls.

5- The results of the primary cultured lymphatic cells are intriguing. Having a long back ground of lymphatic culture cells, I am not 100% convinced that the cells presented here are purely lymphatic cells. Indeed, skin lymphatic cells culture is very challenging and usually it is easier to use a pool of lymph nodes to isolate them. In this article, we don't have any negative controls (secondary antibodies alone but also other cell type to demonstrate the specificity of the antibodies that used in Fig. 3a: PROX1, VEGFR3). I guess that the cells here are a pool of lymphatic + other dermal cells. Authors should provide FACS analysis with lymphatic markers of their isolated population to demonstrate the purity of the cells. This is probably why the Authors are not seeing an increase in AKT phosphorylation.

6- Some important recent works in the PIK3CA field are not referred (Rodrigues Laguna et al JEM 2019, Venot Q et al Nature 2018...) particularly one using a similar genetic approach + rapamycin)

7- The inhibition of VEGF C in their mouse model is absolutely not demonstrated. Authors should provide a demonstration of VEGF C trapping in their mouse model.

Response to the Reviewers NCOMMS-19-23514-T

We thank the Reviewers for their constructive comments that helped us to improve the study. We have revised the manuscript and added several new experiments. In particular, our new experiments conclusively demonstrate that:

- Developmental timing of activation of *Pik3ca* determines the LM phenotype. New data show that 4-OHT administered systemically at different developmental stages (early vs. late embryonic) leads to different LM subtypes (macrocytic vs. microcytic) in the same tissue (dorsal skin of the embryo).
- High PI3K activity induces lymphatic sprouting. We show high PI3K activity (pS6 staining) selectively in lymphangiogenic sprouts *in vivo*. We have further strengthened our previous conclusion that p110 α activation (upon *Pik3ca*^{H1047R} expression) increases and p110 α inhibition (by Rapamycin and VEGF-C blockade) decreases the number of active lymphatic sprouts by providing quantification of our data.
- The migratory phenotype of the *Pik3ca*^{H1047R}-expressing primary LECs is caused by a direct LEC-autonomous effect of increased PI3K signaling and is independent of VEGF-C signaling, even though the subsequent vessel growth *in vivo* is dependent on VEGF-C.

All specific points were addressed as follows (in blue, new and revised figures indicated in bold):

Reviewer #1 (Remarks to the Author):

The paper is well-done and convincing. It is important and has significant clinical translational potential. I believe it should be published in Nature Communications. I don't have any major edits for the manuscript. Minor suggestions include: 1) Figure 1 compares microcytic vs macrocytic lesions histologically but the title of the figure describes both lesions having the same mutation- I would add to the title the specific PIK3CA mutation both lesions have. Also, there is a typo in the legend that should say macrocytic and not microcytic.

Response: Both patients have *PIK3CA*^{H1047R} mutation. We have added this information to the title of Figure 1 legend. The typo has been corrected, also in Supplementary Figure 1.

The second figure would be strengthened if they had clinical photographs of the lymphatic malformations in the mice as well as MRI images to further show they recapitulate the human disease. Lastly, they could add to the discussion that their finding that cervical LMs are more likely to be macrocytic mirrors the human conditions as well (this further strengthens their data).

Response: Lymphatic malformations developing in the ear skin are not visible until after an abrupt lesional bleeding is observed. An example of such a bleeding phenotype is shown in Figure 2e. We also show lesional bleeding in macrocytic LM in Supplementary Figure 2b. This is reminiscent of the pathology in human LM. We agree with the Reviewer that MRI would be useful for further characterization of these lesions, in particular for longitudinal studies monitoring lesion growth and therapy responses. Along the suggestion of the Reviewer, we have added the following sentence to the discussion (page 17):

“Longitudinal magnetic resonance imaging of the LM lesions could provide important insights into the mechanisms of LM growth.”

As suggested, we have also clarified in the discussion that the localization of the macrocystic lesions mimics human pathology (page 15):

*“In mouse skin, embryonic LEC-specific activation of *Pik3ca* led to macrocystic lesions characterized by large cysts localized predominantly to the neck region of the skin, similarly as in human LM patients.”*

Reviewer #2 (Remarks to the Author):

Martinez-Corral et al have investigated lymphatic malformations associated with macrocystic and microcystic lesions. They used human samples (n=5) and generated a mouse model with inducible LEC activation of the main causative mutation of PI3KCA (H104R). The activation of this mutated form has been induced in different mice (using different promoters), at embryonic (E11) and post-natal (P2-5) stages. These approaches led to cysts and lymphatic vessel hypersprouting. The timing of activation determines the type of malformation (macrocystic when induced at an early stage and microcystic at a later stage). The authors provide clear evidence that both microcystic and macrocystic lesions driven by the mutation are dependent to VEGF-C signaling. Importantly, both malformations can be reduced by an inhibitory combination of VEGF-C/VEGFR-3 and PI3K/mTOR pathways.

This is a timely study well conducted and convincing that is providing innovative data with therapeutic potential. There are no major comment.

Specific points:

- page 6: Stainings of PDPN, Prox1 and Lyve-1 should be illustrated (not as data not shown).

Response: As suggested, representative images of micro- and macrocystic malformations stained for PDPN, PROX1 and LYVE1 were added as new panels in **Supplementary Figure 1a**. Text has been revised and is now as follows:

“The endothelium lining the cystic lumens was positive for CD31/PECAM1 (data not shown), PDPN and PROX1 (Supplementary Fig. 1a), confirming lymphatic endothelial identity. LYVE1 (Supplementary Fig. 1a) and VEGFR3 (Fig. 1b, Supplementary Fig. 1b) staining were weak or lacking in the largest cysts, in particular in areas surrounded by α SMA⁺ smooth muscle cells. ... The endothelium of microcystic LMs was positive for VEGFR3 (Fig. 1b), PDPN and PROX1 (Supplementary Fig. 1a). LYVE1 staining was patchy, similarly as in macrocystic LMs (Supplementary Fig. 1a).”

- Figure 3: Zipper junctions are illustrated as a hallmark of lymphangiogenic vessels. However, a comparison with WT tissue should be provided.

Response: Transformation of button junctions of normal lymphatic capillaries into zippers in the actively growing vessels was first described during inflammation-induced lymphangiogenesis by Donald McDonald's group^{1,2}. Zippers were later reported also in actively sprouting embryonic lymphatic vessels³. In agreement with previous findings⁴, we found that the fully remodeled dermal lymphatic vasculature of the ear of a 3-week old wild type mouse has only a few active lymphatic sprouts that show a typical spiky appearance (new **Figure 3b**). At 5 weeks of age, no active sprouts were observed (new **Figure 5f**). At these timepoints, the blind ends of normal lymphatic capillaries instead have a blunted morphology and their endothelial lining shows button junctions. This is now shown in **Figure 3d** (previous Figure 3a).

- Faster wound closure is detected under basal conditions. The dependence to VEGF-C should be documented or at least discussed.
- Does the mutation activation in primary LEC affect VEGFR-3 and VEGFR-2 levels and activation by VEGF-C? Do cells respond equally to WT VEGF-C and mutated VEGFC156S? This could be tested, at least in the scratch assay.

Response: Following the Reviewer's suggestion, we have included analysis of scratch wound closure in the presence of VEGF-C. New **Supplementary Fig. 5b** summarizes wound closure rates calculated as area under the curve (AUC) from the relative wound density. We observed no further increase in the speed of wound closure in the *Pik3ca*^{H1047R}-expressing primary LECs in the presence of VEGF-C. Therefore we did not test separately the effect of the VEGFR3-specific VEGF-C C156S. This finding suggests that the migratory phenotype of *Pik3ca*^{H1047R}-expressing LECs is not, at least *in vitro*, dependent on VEGF-C but instead represents a cell-autonomous effect of oncogenic PI3K activity on the LECs. It should be noted, however, that the wounding assay induces directional collective migration upon the scratching-induced relief of contact inhibition. However, this cannot be used to assess the importance of VEGF-C for directional growth of lymphatic sprouts *in vivo*.

Western blot analysis of control and *Pik3ca*^{H1047R}-expressing primary LECs indicated no differences in VEGFR2 or VEGFR3 levels (new **Supplementary Fig. 6e, f**), suggesting that the upregulation observed *in vivo* is not a direct effect of increased PI3K signaling.

Based on our previous and new data above, we propose that the migratory phenotype of the *Pik3ca*^{H1047R}-expressing primary LECs is due to a direct LEC-autonomous effect of increased PI3K signaling. We have clarified this in the discussion (page 18):

"In addition to the cell-autonomous migration-promoting effect of p110α on LECs observed in vitro, lymphangiogenic VEGF-C/VEGFR3 signaling in the LM lesion was found to significantly contribute to the further vascular growth of the lesions in vivo."

Reviewer #3 (Remarks to the Author):

In this article, Dr. Martinez Corral and coauthors explore the role played by PIK3CA mutation in lymphatic cells. Using different mouse models, they observed different subgroup of lymphatic malformation and identified a dual therapy as potential new therapeutic opportunity for patients. The paper is well written, easy to read but I have some important concerns regarding the interpretation.

1-The first conclusion is potentially wrong. Indeed, it is difficult to say that the occurrence of an early mutation during embryological development is leading to macrocystic lesion and late mutation to microcystic lymphatic malformation. In the first case the authors have injected 4 OH tamoxifen to pregnant mice (inducing indeed a systemic recombination in deep lymphatic cells) and on the other hand they applied locally 4OHT which is inducing the mutation only in superficial skin lymphatic vessels. The message is indeed completely different and overstated. By the way, this technical part is not well described and I was confused when I was reading the results.

Response: We agree with the Reviewer that the effect of 4-OHT (systemic vs. local recombination) in the different models used in the study was not explained in sufficient detail. We now provide additional data in **Supplementary Table 1**, new **Supplementary Figure 3a, b**, and new **Supplementary Figure 4a-d**. Specifically, we strengthen our conclusion that the developmental timing of activation of *Pik3ca* determines the LM phenotype by showing the following evidence:

- 1) Administration of 4-OHT to pregnant females by intraperitoneal injection (i.e. systemically) at a later developmental stage (E14) led to lymphatic vessel hyperbranching in the embryonic skin (new **Supplementary Figure 2d, e**). This is reminiscent of microcystic lesions formed upon postnatal induction of *Pik3ca*^{H1047R} expression (Figure 2e), but differs from induction of the macrocystic lesions induced by an early embryonic 4-OHT administration (Figure 2b, Supplementary Figure 2b). Thus, 4-OHT administered systemically at different developmental stages (early vs. late embryonic) leads to different LM subtypes in the same tissue (dorsal skin of the embryo).
- 2) Topical application of 100 µg of 4-OHT to one ear leads to systemic recombination in the *Vegfr3-CreER*^{T2} model, as demonstrated by Cre-mediated recombination of the *R26-mTmG* reporter (new **Supplementary Figure 4a, b**). Notably, the recombination rate in both the treated and untreated ears is high, likely reflecting strong VEGFR3 expression (translating into efficient Cre recombination) in dermal lymphatic vessels, while recombination in other tissues such as the intestine and diaphragm is mosaic (new **Supplementary Figure 4b**). Furthermore, quantification of lymphatic vessel branching in the *Pik3ca*^{H1047R}; *Vegfr3-CreER*^{T2} mice showed no difference between 4-OHT-treated and untreated ears from the same mice (new **Supplementary Figure 4c, d**).
- 3) Topical application of a low dose (0.5-2 µg) of 4-OHT to one ear results in locally restricted (mosaic) recombination in the 4-OHT treated ear in the *Prox1-CreER*^{T2} model. Lymphatic lesions were observed in 14/14 treated ears of mice developing lesions, but only in 1/14 untreated ears of the same mice (new **Supplementary Figure 3b**). Low incidence of tamoxifen-independent recombination⁵ cannot be excluded as a cause of lesion formation in the untreated ear of the only mouse that had recombination.

2-Several mandatory controls are missing: Fig. 3a (LYVE1 pS6 staining in normal lymphatic vessels at baseline...), Fig. 4A...

Response: We have stained normal lymphatic vasculature for pS6, and observe signal selectively in the actively growing embryonic vessels and in the tips of active sprouts in particular, but not in the quiescent vasculature of the ear (new **Figure 3c**). This is in agreement with our previous finding that in the *Pik3ca* mutant mice, only active lymphatic sprouts, characterized by a typical spiky morphology, show high pS6 staining (Figure 3c (previously Figure 3a)).

To further strengthen the link between high PI3K activity and induction of lymphatic sprouting in the *Pik3ca* mutant mice, we have quantified the number of vessel ends and their morphology in the lymphatic vasculature of control and *Pik3ca* mutant ears. In agreement with previous findings⁴, we found that the fully remodeled dermal lymphatic vasculature of the ear of a 3-week old wild type mouse has only a few active lymphatic sprouts characterized by a typical spiky appearance (new **Figure 3b**). Instead, the blind ends of normal lymphatic capillaries show a blunted morphology. We observed an increase in the number of vessel ends in *Pik3ca* mutant mice, and the majority of these showed a spiky morphology, representing new lymphangiogenic vessel sprouts (new **Figure 3b**). This parameter was so informative that we have added new quantification of the morphology of vessel ends in mice treated with Rapamycin and/or AAV-sR3. This data confirms a decrease in the spiky (active) vessel sprouts after the combination therapy (new **Figure 5f**).

3-I am concerned about the WB. Why the authors do not detect any AKT phosphorylation in their primary cultured cells? It is very strange and unusual. Furthermore, the IF in Fig. 3A shows activation of mTORC1, a lower step in the cascade. IF of PAKT but also WB of PS6 are requested. Of note, this

signaling pathway is extremely sensitive to starving (mandatory before to interpret any in vitro and in vivo data).

Response: In Figure 3h (previous Figure 3d), basal AKT phosphorylation is low, because the experiment was done under serum starvation, to allow assessment of the specific effect of VEGF-C stimulation on pAKT levels (as noted by the Reviewer, this is necessary). For clarity, serum starvation has now been stated in Figure 3h. The exposure time during image capturing will determine the absolute signal intensities of the bands, and therefore Western blot analysis can only demonstrate comparison of the relative levels of signal intensity between the samples. To allow reliable quantification of signal intensities, oversaturation of the signal has to be avoided; in this case optimal exposure time for quantification of the VEGF-C stimulated samples resulted in only a weak band that was observed in the unstimulated control cells. Below is a longer exposure of the blot (Figure 3h) for the Reviewer, to show the presence of low basal AKT phosphorylation in the unstimulated serum-starved control LECs (**Figure 1 for Reviewers**).

Figure 1 for Reviewers. Western blot analysis of phospho-AKT in serum-starved primary dermal LECs from *Pik3ca*^{H1047R}; *Vegfr3-CreER*^{T2} mice treated *in vitro* with 4-OHT and/or VEGF-C for indicated times. A longer exposure of pAKT-Ser473 is included as the middle panel.

We also provide new Western blot data showing pAKT levels in control and *Pik3ca*^{H1047R}-expressing primary LECs grown in complete growth medium (new **Figure 3g**). Basal pAKT levels are higher in control cells, but a robust increase in phosphorylation is nevertheless detected in the mutant cells. We further show that in addition to phosphorylation of Ser473 (that is targeted by both mTORC2 and PI3K), Thr308 of AKT (targeted specifically by the PI3K pathway) is also increased (new panel was added in **Figure 3h**).

To the best of our knowledge, antibodies that allow visualizing pAKT by whole-mount immunofluorescence are not available, and our attempts to stain pAKT were not successful. We followed the Reviewer's suggestion to visualize pS6 by Western blot in cultured LECs. Under the various growth conditions tested (full growth medium vs. starvation, low vs. high confluency), we were not able to detect an increase in pS6-Ser240/244 levels in the *Pik3ca*^{H1047R}-expressing LECs compared to controls (**Figure 2a, b for Reviewers**). This may not be entirely surprising, considering that pS6 phosphorylation occurs in response to a wide variety of stimuli⁶, including activation by the Ras/Raf/MEK/ERK pathway that was relatively strong and was not affected by expression of *Pik3ca*^{H1047R} in the cultured mouse LECs (**Figure 2a, b for Reviewers**). In agreement with these data, Castel et al. demonstrated only a modest induction of pS6 despite strong AKT activation in cultured HUVECs expressing *PIK3CA*^{H1047R} (ref⁷, the relevant blot reproduced in **Figure 2c for Reviewers** for comparison). Due to the relatively high baseline phosphorylation of S6 in cultured LECs, even under serum starvation (**Figure 2b for Reviewers**), the weak increase caused by PI3K activation may thus be masked. Nevertheless, pS6 has previously been used to report increased PI3K activity in *Pik3ca*^{H1047R}-

expressing retinal endothelial cells *in vivo*⁸. Consistent with this, we find that pS6 is only detected in the actively growing embryonic vessels (new **Figure 3c**) and in new sprouts induced upon p110 α activation, but not in the quiescent vasculature of the ear (new **Figure 3c**).

Figure 2 for Reviewers. Western blot analysis of phospho-S6-Ser240/244 in primary dermal LECs from *Pik3ca*^{H1047R};*Vegfr3-CreER*^{T2} mice treated *in vitro* with 4-OHT and cultured either in full medium (**a**) or stimulated with VEGF-C after serum starvation (**b**). Cell confluence (high/low) and time of VEGF-C stimulation are indicated. Panel **c** showing pAKT and pS6 levels in HUVECs expressing the wild type *PIK3CA* or mutant *PIK3CA*^{H1047R} is reproduced from⁷. Red boxes indicate the blots that are shown in new Figure 3 g, h.

4- Fig. 4a: VEGFR2 staining seems to be nuclear. Again, no controls.

Response: We provide co-staining of dermal lymphatic vessels for VEGFR2 and PROX1 to demonstrate that the punctuate VEGFR2 staining is not nuclear (**Figure 3 for Reviewers**). The punctuate perinuclear staining likely represents the reported localization of VEGFR2 in the Golgi and intracellular vesicles, reflecting presence of newly synthesized protein and its vesicular trafficking (for a review see^{9,10}). Of note, we have previously validated the specificity of the VEGFR2 antibody used in the study, by demonstrating lack of staining in *Vegfr2*-deleted mice¹¹.

Figure 3 for Reviewers. Whole-mount immunofluorescence staining of mouse ear skin showing punctate perinuclear staining of VEGFR2 in lymphatic vessels (arrows).

As suggested, we have added controls to **Figure 4a** by showing images of lymphatic vessels of wild type mice stained for VEGFR2, VEGFR3 and NRP2.

5- The results of the primary cultured lymphatic cells are intriguing. Having a long back ground of lymphatic culture cells, I am not 100% convinced that the cells presented here are purely lymphatic cells. Indeed, skin lymphatic cells culture is very challenging and usually it is easier to use a pool of

lymph nodes to isolate them. In this article, we don't have any negative controls (secondary antibodies alone but also other cell type to demonstrate the specificity of the antibodies that used in Fig. 3a: PROX1, VEGFR3). I guess that the cells here are a pool of lymphatic + other dermal cells. Authors should provide FACS analysis with lymphatic markers of their isolated population to demonstrate the purity of the cells. This is probably why the Authors are not seeing an increase in AKT phosphorylation.

Response: We have previously demonstrated by immunofluorescence and FACS analysis that our cell isolation method, which includes sequential selection of PECAM1⁺ ECs and LYVE1⁺ LECs, gives a highly pure population of dermal LECs¹². LEC identity in such cultures was previously shown by staining for PROX1 and VE-cadherin, as well as by assessing the expression of a genetic GFP reporter driven by the *Vegfr3* promoter (*Vegfr3-CreER^{T2};R26-mTmG*)¹² that we use here to induce Cre-mediated activation of *Pik3ca^{H1047R}* expression *in vitro*. We have similarly controlled and now quantified (new **Supplementary Figure 5a**) the purity of each LEC batch used for our experiments in this study by staining for PROX1 and VE-cadherin. All antibodies have been used and validated in previous studies. For example, the specificity of the PROX1 antibody was previously demonstrated in cultured human dermal ECs, with staining observed specifically in the nucleus of LECs but not BECs¹³. Staining of dermal mouse ECs, consisting of a mixed population of BECs and LECs, also shows PROX1 staining in only a proportion of VE-cadherin⁺ cells (**Figure 4 for Reviewers**, also evident in **Supplementary Figure 5a**, red asterisks). Quantifications shows that the cells used in this study are all VE-cad⁺ (100 ± 0 %), while (89.7 ± 4.3 %) are PROX1⁺ (*n*=1386 cells in total from 15 experimental conditions and 2 independent batches of cell isolation) (**Supplementary Figure 5a**). Thus, they are LECs with a minor (maximally 10%) contamination with BEC. A strength of our approach is that the same batch of cells provides both control LECs (-4-OHT) and *Pik3ca^{H1047R}* LECs (+4-OHT), and thus phenotypic differences between the two are not due to batch-to-batch differences (e.g. in the % of contaminating BEC).

PECAM1-bead selected dermal EC (BEC and LEC) from a *Pik3ca^{H1047R}; Vegfr3-CreER^{T2}* mouse

Figure 4 for Reviewers. Immunofluorescence of PECAM1-bead selected dermal mouse ECs consisting of a mixture of BECs and LECs. Note junctional VE-cadherin staining in all cells. A subset of VE-cad⁺ cells are positive for the LEC-marker PROX1 (arrows, LEC) while some VE-cad⁺ cells are PROX1⁻ (arrowheads, BEC).

We would like to refer the Reviewer to Figure 3 panels h, i (previous Figure 3d, e), which show that the cells respond robustly to VEGF-C stimulation by increased AKT phosphorylation (about 10-fold increase in pAKT signal was observed in control LECs (Figure 3i)).

Considering the known organ-specific features of ECs^{14,15}, including the unique properties of lymph node LECs^{16,17}, as well as the vascular bed-specific manifestation of *PIK3CA*-driven vascular malformations¹⁸, LECs isolated from lymph nodes may not be relevant for studying *Pik3ca*-driven LM that we study here in the context of skin.

6- Some important recent works in the PIK3CA field are not referred (Rodrigues Laguna et al JEM 2019, Venot Q et al Nature 2018...) particularly one using a similar genetic approach + rapamycin)

Response: Thank you for this reminder. We previously referred to Rodrigues-Laguna et al. only in the introduction, but now mention both papers also in the discussion. New text is indicated in bold:

Page 16: “LEC-specific activation of *Pik3ca*^{H1047R} expression at three weeks of age led to a generalized, rather than localized response characterized by progressive lymphatic vascular hyperplasia. **This is in agreement with a previous study modelling generalized lymphatic anomaly (GLA) in mice upon LEC-specific induction of *Pik3ca*^{H1047R} at four weeks of age¹⁹.**”

Page 17: “Consistent with the clinical data **and previous studies on a mouse model of *Pik3ca*-driven generalized lymphatic anomaly¹⁹**, we found that Rapamycin inhibited lymphatic vascular growth in the experimental model of *Pik3ca*^{H1047R}-driven microcystic LM.”

Page 18: “Interestingly, Alpelisib (BYL719), a specific PIK3CA inhibitor, has shown a higher efficacy than Rapamycin in the treatment of PIK3CA-related overgrowth syndromes (PROS) in mice and humans²⁰. Alpelisib improved organ abnormalities and dysfunction, also in the vasculature, with minimal side effects. Future studies should address the efficacy of Alpelisib in patients with LM.”

7- The inhibition of VEGF C in their mouse model is absolutely not demonstrated. Authors should provide a demonstration of VEGF C trapping in their mouse model.

Response: VEGF-C trap, consisting of the ligand binding domain of VEGFR3 fused to the IgG Fc domain, has been extensively used to inhibit VEGF-C activity *in vivo* (e.g. ^{21–25}). In this study, we used the AAV-encoded VEGF-C trap (AAV-VEGFR3₁₋₄-Ig) from Prof. Kari Alitalo’s laboratory that has been validated and used in a number of published studies (e.g. ^{23–25}), including ours¹¹. The dose and the route of administration are in accordance with previous studies.

To directly address the Reviewers’ request to demonstrate VEGF-C trapping in the specific experimental conditions of this study, we analyzed the sera of AAV-treated *Pik3ca* mutant mice by Western blot, as described²⁶. Soluble VEGFR3₁₋₄-Ig (or the control VEGFR3₄₋₇-Ig) protein was detected in the serum 3 weeks after intraperitoneal AAV administration at a concentration of 121 ± 30 ng/μl (n=5) (new **Supplementary Figure 7a**), which was previously shown to effectively neutralize lymphangiogenic signaling²¹. We further confirmed that this amount of protein present in the serum was sufficient to block VEGF-C -induced lymphangiogenesis in the ear skin after intradermal injection of AAV vectors encoding VEGF-C (new **Supplementary Figure 7b**).

References

1. Baluk, P. *et al.* Functionally specialized junctions between endothelial cells of lymphatic vessels. *J. Exp. Med.* **204**, 2349–2362 (2007).
2. Yao, L.-C., Baluk, P., Srinivasan, R. S., Oliver, G. & McDonald, D. M. Plasticity of button-like junctions in the endothelium of airway lymphatics in development and inflammation. *Am. J. Pathol.* **180**, 2561–2575 (2012).
3. Zheng, W. *et al.* Angiopoietin 2 regulates the transformation and integrity of lymphatic endothelial cell junctions. *Genes Dev.* **28**, 1592–1603 (2014).
4. Bernier-Latmani, J. *et al.* DLL4 promotes continuous adult intestinal lacteal regeneration and dietary fat transport. *J. Clin. Invest.* **125**, 4572–4586 (2015).

5. Álvarez-Aznar, A. *et al.* Tamoxifen-independent recombination of reporter genes limits lineage tracing and mosaic analysis using CreERT2 lines. *Transgenic Res.* (2019) doi:10.1007/s11248-019-00177-8.
6. Meyuhas, O. Ribosomal Protein S6 Phosphorylation: Four Decades of Research. *Int. Rev. Cell Mol. Biol.* **320**, 41–73 (2015).
7. Castel, P. *et al.* Somatic PIK3CA mutations as a driver of sporadic venous malformations. *Sci. Transl. Med.* **8**, 332ra42 (2016).
8. Angulo-Urarte, A. *et al.* Endothelial cell rearrangements during vascular patterning require PI3-kinase-mediated inhibition of actomyosin contractility. *Nat. Commun.* **9**, 4826 (2018).
9. Simons, M., Gordon, E. & Claesson-Welsh, L. Mechanisms and regulation of endothelial VEGF receptor signalling. *Nat. Rev. Mol. Cell Biol.* **17**, 611–625 (2016).
10. Secker, G. A. & Harvey, N. L. VEGFR signaling during lymphatic vascular development: From progenitor cells to functional vessels. *Dev. Dyn. Off. Publ. Am. Assoc. Anat.* **244**, 323–331 (2015).
11. Zhang, Y. *et al.* Heterogeneity in VEGFR3 levels drives lymphatic vessel hyperplasia through cell-autonomous and non-cell-autonomous mechanisms. *Nat. Commun.* **9**, 1296 (2018).
12. Martínez-Corral, I. *et al.* Vegfr3-CreER (T2) mouse, a new genetic tool for targeting the lymphatic system. *Angiogenesis* (2016) doi:10.1007/s10456-016-9505-x.
13. Petrova, T. V. *et al.* Lymphatic endothelial reprogramming of vascular endothelial cells by the Prox-1 homeobox transcription factor. *EMBO J.* **21**, 4593–4599 (2002).
14. Petrova, T. V. & Koh, G. Y. Organ-specific lymphatic vasculature: From development to pathophysiology. *J. Exp. Med.* **215**, 35–49 (2018).
15. Potente, M. & Mäkinen, T. Vascular heterogeneity and specialization in development and disease. *Nat. Rev. Mol. Cell Biol.* **18**, 477–494 (2017).
16. Takeda, A. *et al.* Single-Cell Survey of Human Lymphatics Unveils Marked Endothelial Cell Heterogeneity and Mechanisms of Homing for Neutrophils. *Immunity* **51**, 561-572.e5 (2019).
17. Berendam, S. J. *et al.* Comparative Transcriptomic Analysis Identifies a Range of Immunologically Related Functional Elaborations of Lymph Node Associated Lymphatic and Blood Endothelial Cells. *Front. Immunol.* **10**, 816 (2019).
18. Limaye, N. *et al.* Somatic Activating PIK3CA Mutations Cause Venous Malformation. *Am. J. Hum. Genet.* **97**, 914–921 (2015).
19. Rodríguez-Laguna, L. *et al.* Somatic activating mutations in PIK3CA cause generalized lymphatic anomaly. *J. Exp. Med.* **216**, 407–418 (2019).
20. Venot, Q. *et al.* Targeted therapy in patients with PIK3CA-related overgrowth syndrome. *Nature* **558**, 540–546 (2018).
21. Mäkinen, T. *et al.* Inhibition of lymphangiogenesis with resulting lymphedema in transgenic mice expressing soluble VEGF receptor-3. *Nat. Med.* **7**, 199–205 (2001).
22. Karpanen, T. *et al.* Lymphangiogenic growth factor responsiveness is modulated by postnatal lymphatic vessel maturation. *Am. J. Pathol.* **169**, 708–718 (2006).
23. Alitalo, A. K. *et al.* VEGF-C and VEGF-D blockade inhibits inflammatory skin carcinogenesis. *Cancer Res.* **73**, 4212–4221 (2013).
24. Fang, S. *et al.* Critical requirement of VEGF-C in transition to fetal erythropoiesis. *Blood* **128**, 710–720 (2016).
25. Antila, S. *et al.* Development and plasticity of meningeal lymphatic vessels. *J. Exp. Med.* **214**, 3645–3667 (2017).
26. Karaman, S., Nurmi, H., Antila, S. & Alitalo, K. Stimulation and Inhibition of Lymphangiogenesis Via Adeno-Associated Viral Gene Delivery. *Methods Mol. Biol. Clifton NJ* **1846**, 291–300 (2018).

Reviewers' Comments:

Reviewer #1:

Remarks to the Author:

The authors have adequately revised the manuscript. The paper is well-done, convincing, and has high impact. It should be published in Nature Communications.

Reviewer #2:

Remarks to the Author:

The authors have answered to all comments.

Reviewer #3:

Remarks to the Author:

The authors have been highly responsive to the comments raised during previous review which has led to an improved version of the manuscript. I have no further comments in reference to this revised version .

NCOMMS-19-23514A

Reviewers did not raise any further comments.

REVIEWERS' COMMENTS:

Reviewer #1 (Remarks to the Author):

The authors have adequately revised the manuscript. The paper is well-done, convincing, and has high impact. It should be published in Nature Communications.

Reviewer #2 (Remarks to the Author):

The authors have answered to all comments.

Reviewer #3 (Remarks to the Author):

The authors have been highly responsive to the comments raised during previous review which has let to an improved version of the manuscript. I have no further comments in reference to this revised version .